# Think Right: Learning to Mitigate Under-Over Thinking via Adaptive, Attentive Compression

## Abstract

Recent thinking models are capable of solving complex reasoning tasks by scaling test-time compute across various domains, but this scaling must be allocated in line with task difficulty. On one hand, short reasoning (underthinking) leads to errors on harder problems that require extended reasoning steps; but, excessively long reasoning (overthinking) can be token-inefficient by generating unnecessary steps even after reaching a correct intermediate solution. We refer to this as **under-adaptivity**, where the model fails to modulate its response length appropriately given problems of varying difficulty. To address under-adaptivity and strike a balance between under- and overthinking, we propose TRAAC (**T**hink **R**ight with **A**daptive, **A**ttentive **C**ompression), an online post-training RL method that leverages the model's self-attention over a long reasoning trajectory to identify important steps and prune redundant ones. TRAAC also estimates difficulty and incorporates into training rewards, thereby learning to allocate reasoning budget commensurate with example difficulty. Our approach improves accuracy, reduces reasoning steps, and enables adaptive thinking compared to base models and other RL baselines. Across a variety of tasks (AIME, AMC, GPQA-D, BBEH), TRAAC (Qwen3-4B) achieves an average absolute accuracy gain of 8.4% with a relative reduction in reasoning length of 36.8% compared to the base model, and a 7.9% accuracy gain paired with a 29.4% length drop compared to the best RL baseline. TRAAC also shows strong generalization: although the models are trained on math datasets, they show accuracy and efficiency gains on out-of-distribution non-math datasets like GPQA-D, BBEH, and OptimalThinkingBench. Our analysis further verifies that TRAAC provides fine-grained adjustments to thinking budget based on difficulty and that a combination of task-difficulty calibration and attention-based compression yields gains across diverse tasks.[1]

## 1 Introduction

Recent advancements in thinking models have enabled language models to solve complex reasoning tasks (DeepSeek-AI et al., 2025; OpenAI et al., 2024; Team, 2025). These models extend the chain-of-thought (CoT; Wei et al., 2023) paradigm with online reinforcement learning (RL; Shao et al., 2024), allowing them to refine intermediate solutions as well as sequentially scaling the number of tokens (i.e., compute) to arrive at the final answer. While such approaches show strong promise for harder problems in domains like mathematics, programming, and logical puzzles (Xie et al., 2025; Chen et al., 2025), their accuracy and utility remain capped by a failure to regulate their reasoning length. On one hand, *underthinking* arises when models terminate too early on harder problems, yielding an incorrect final answer. On the other hand, *overthinking* occurs when models think excessively for simpler tasks, inflating test-time computation (Marjanović et al., 2025; Wu et al., 2025; Cuadron et al., 2025), and reducing efficiency. This highlights the need for adaptive thinking (Saha et al., 2025; Chen et al., 2024; Snell et al., 2024; Aggarwal & Welleck, 2025), where models dynamically allocate thinking based on difficulty.

We refer to the phenomenon of models misallocating thinking budget – illustrated in Fig. 1 – as *under-adaptivity*. Addressing under-adaptivity is crucial for improving both performance and efficiency of long-thinking models, as dynamic reasoning effort allocation can enable better reason-

---

[1]Codebase available in the supplementary, and will be released publicly upon acceptance.

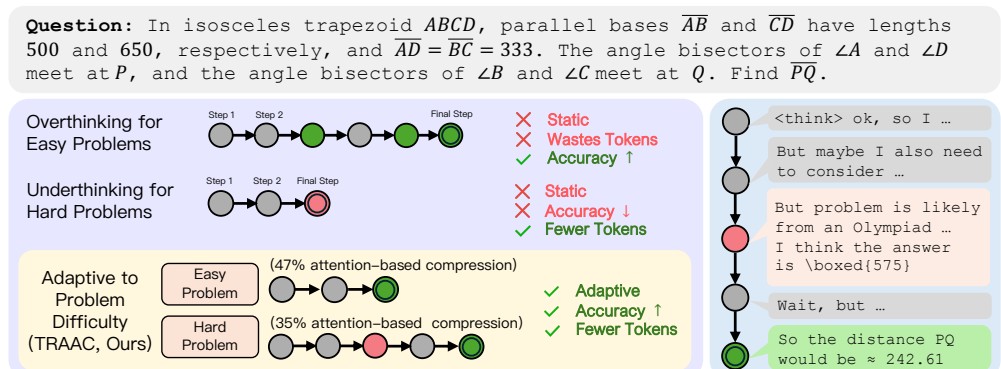

Figure 1: Overthinking on easy problems wastes tokens despite being able to maintain decent accuracy. On the other hand, underthinking on hard problems saves token budgets but fails to maintain accuracy. TRAAC addresses this trade-off by adapting to problem difficulty (estimated during training), via attention-based compression and, enabling intelligent resource allocation while improving both accuracy and efficiency.

ing exploration in harder problems, while avoiding wasteful computation on problems requiring minimal reasoning. Prior work has generally addressed the upper end of under-adaptivity, i.e., improving thinking efficiency. These works employ supervised fine-tuning on compressed CoT (Xia et al., 2025), using user control signals such as early stopping during inference (Muennighoff et al., 2025), or RL methods with length penalties (Arora & Zanette, 2025; Hou et al., 2025). Other more adaptive work has employed budget-aware reward shaping with a binary choice between thinking or not thinking (Zhang et al., 2025b). While such work can reduce token usage, its performance is typically bounded by the accuracy of the underlying model being trained, and often trades performance for efficiency. Our work aims to beat this trade-off and improve both efficiency and accuracy by providing finer-grained feedback through difficulty-adaptive compression, where the degree of compression is dynamically adapted to task difficulty to address under-adaptivity.

To address these gaps, we introduce TRAAC (**T**hink **R**ight with **A**daptive, **A**ttentive **C**ompression), a GRPO-based (Shao et al., 2024) post-training method that incorporates an **online, difficulty-adaptive, attention-based compression module** to adaptively prune the reasoning trajectory (an entire chain in Fig. 1) based on estimated task difficulty. Our method teaches the model to compress the context that it should pay attention to, such that it contains only relevant material without getting distracted or skewed in wrong directions (Weston & Sukhbaatar, 2023). Specifically, we compute the attention score averaged across layers and heads of the model for each reasoning step (illustrated as nodes in Fig. 1 (right)) from the `</think>` token and *compress* reasoning steps that are *least attended to*, based on the assumption that these are the least important tokens contributing to the final answer. During online training, the level of attention-compression is determined by task difficulty, as estimated by the pass rate during GRPO rollout, making the model more adaptive. For harder problems, TRAAC maintains a low compression rate, allowing the model to extend its reasoning trajectory, which increases the likelihood of reaching the correct final answer. For easier problems, it applies a higher compression rate to aggressively compress once the correct final answer is reached.

We evaluate TRAAC on two strong off-the-shelf reasoning models, Qwen3-4B (Team, 2025) and Deepseek-Qwen-7B (DeepSeek-AI et al., 2025), across multiple benchmarks: AMC (AMC, 2023), AIME (AIME, 2024), GPQA-Diamond (Rein et al., 2023), BBEH (Big Bench Extra Hard; Kazemi et al., 2025), and OptimalThinkingBench (Aggarwal et al., 2025). Our experiments demonstrate that TRAAC consistently adapts to problem difficulty, yielding improvements in efficiency on simple tasks and stronger accuracy on complex tasks. Averaged across AMC, AIME, GPQA-D, and BBEH, TRAAC (Qwen3-4B) achieves an average absolute improvement of 8.4% in accuracy while a relative reduction in reasoning length by 36.8% compared to the base model. When compared to the next-best performing baseline, AdaptThink (Zhang et al., 2025b), we achieve an average accuracy improvement of 7.9% and 29.4% efficiency gain. We test TRAAC on OptimalThinkingBench (Aggarwal et al., 2025), and find TRAAC improves by 7.36 points on Qwen3-4B and 12.55 points on Deepseek-Qwen-7B over the base model according to Aggarwal et al. (2025)'s F1 metric – designed to measure *both performance and efficiency*. Moreover, TRAAC is trained on a math-specific dataset;

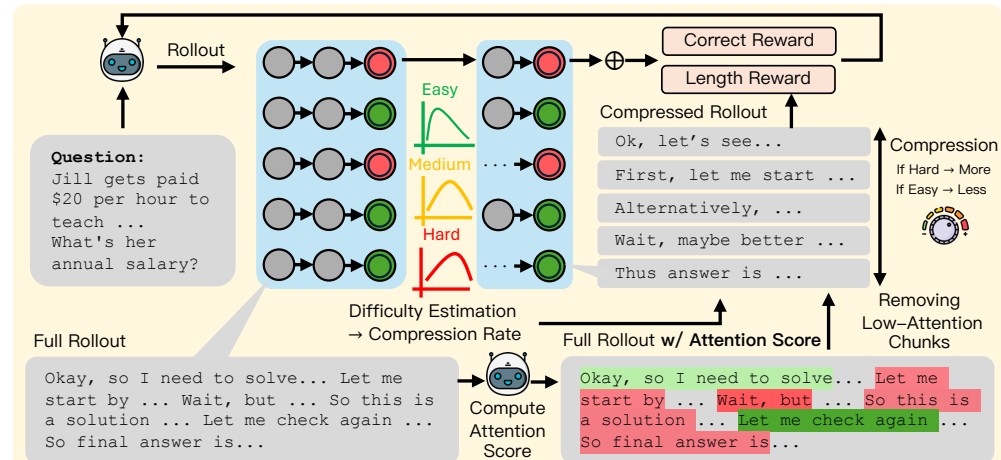

Figure 2: Overview of TRAAC. Given a problem, the model first generates $N$ rollouts, and the pass rate of these rollouts is used to estimate the problem's difficulty (easy, medium, or hard). Next, the generated reasoning is fed back into the model, which is asked to compute the attention score of each reasoning token from `</think>`. During this attention-based compression step, we remove steps with lower scores. The degree of removal is determined by the estimated difficulty: easier problems undergo more aggressive compression. Finally, we compute the correctness and length rewards using the compressed reasoning trajectory, and these rewards are used to update the policy.

evaluation on various benchmarks such as GPQA-D, BBEH, OverthinkingBench, and Underthink-ingBench shows generalizability performance. Among these OOD tasks, TRAAC shows an average improvement of 3% on Qwen3-4B, with a maximum improvement of 6.8% on UnderthinkBench, along with an average 40% reduction in response length across OOD tasks. Our analysis and ablations demonstrate that through difficulty level calibration, TRAAC learns to dynamically adjust its compression ratio – with lower compression on difficult tasks and higher compression on easier ones, which translates into performance gains across diverse difficulty tasks.

## 2   TRAAC: T̲HINK R̲IGHT WITH A̲DAPTIVE A̲TTENTIVE C̲OMPRESSION

In this section, we introduce our proposed TRAAC method in detail (also shown in Fig. 2). It is designed to mitigate under-adaptivity, which leads to resource misallocation during test-time. The main challenge lies in the efficient identification of low-importance tokens and making the attention-based compression adaptive to the task's difficulty. To this end, TRAAC employs an attention-based compression module that calibrates its degree of compression based on estimated task difficulty and prunes unnecessary reasoning steps while preserving essential information.

### 2.1   PROBLEM FORMULATION IN TRAAC

TRAAC is based on Group Reward Policy Optimization (GRPO; Shao et al., 2024), which is an online reinforcement learning (RL) framework that extends Proximal Policy Optimization (Schulman et al., 2017) by eliminating the critic and instead estimating the baseline from a group of sampled responses. Let $\pi_\theta$ denote the policy model and $q$ the input query. Given $q$, the model generates an output $y = \mathrm{cat}(r, a)$ where $\mathrm{cat}$ is the concatenate function, $r$ is the complete reasoning trajectory, and $a$ is the final answer, separated by the delimiter `</think>`. An attention-based compression module $\mathcal{C}$ (described below) produces a compressed reasoning trajectory: $r_{\mathrm{comp}} = \mathcal{C}(r)$. At each training step, the model generates $N$ rollouts, $\{y^i\}_{i=1}^N$, where each rollout $y^i = \mathrm{cat}(r^i, a^i)$ (see "rollout" arrow in Fig. 2). The advantage of each rollout is estimated using the standard GRPO objective (details in Appendix A.9). The task difficulty d is estimated from these rollouts as the proportion of correct answers among the $N$ samples (Zhang & Zuo, 2025; Huang et al., 2025). We show this in Fig. 2 by classifying a problem to easy, medium or hard based on d. Task difficulty d is then used to (i) modulate the compression ratio applied to the reasoning trajectory $r$, and (ii)

assign rewards to each rollout. The answer is regenerated based on the compressed trajectory and the advantage is estimated using both the original rollouts and their compressed counterparts.

## 2.2 ADAPTIVE, ATTENTIVE COMPRESSION MODULE

The goal of the compression module is to identify and remove redundant reasoning steps by evaluating attention scores assigned to each token.

**Attention-Based Compression.** To calculate the attention score assigned to each token, we pass the reasoning trajectory $r$ (full rollout in Fig. 2) through the initial policy model. As compared to other compression-based methods (Cheng et al., 2025; Lu et al., 2025), TRAAC does not rely on external models for annotating reasoning steps. To segment the reasoning trajectory $r$ into reasoning steps, we split it at special control tokens such as *"wait"*, *"alternative"*, *"Let me think again"*, etc. (complete list Appendix A.8.2). For the current thinking models, </think> marks the end of a reasoning trajectory, followed by the final answer. Choi et al. (2025) show that </think> attends to key reasoning steps that contain crucial information for deriving the final answer, therefore, for each token $t_j$ in the reasoning steps, its importance score is defined as the aggregated attention from the delimiter </think> across all layers and heads:

$$s_j = \frac{1}{LH} \sum_{\ell=1}^{L} \sum_{h=1}^{H} \alpha_{\texttt{</think>} \to t_j}^{(\ell, h)},$$

where $L$ is the number of layers, $H$ is the number of heads per layer, and $\alpha_{\texttt{</think>} \to t_j}^{(\ell, h)}$ is the attention weight from </think> to token $t_j$ in head $h$ of layer $\ell$. Before computing the attention score of each token, consistent with prior work (Muennighoff et al., 2025; Choi et al., 2025), we also append an auxiliary prompt at the end of the reasoning trajectory (see Appendix A.8.1). This encourages the model to distill the reasoning process into its most salient steps, thereby enabling the delimiter token </think> to attend to the most informative parts of the reasoning trajectory (highlighted in green). As shown in Fig. 2 (bottom-right), the model assigns low attention scores to reasoning steps that do not contribute to the final correct answer (highlighted in red), effectively pruning unnecessary cyclic self-corrections and verification loops. Finally, the importance score of a reasoning step $C_k$, consisting of tokens $\{t_j\}_{j \in C_k}$, is then computed as the mean of its token-level scores: $s_{C_k} = \frac{1}{|C_k|} \sum_{j \in C_k} s_j$. Steps with lower importance scores are pruned, yielding the compressed reasoning trajectory $r_{\text{comp}}^i$.

**Difficulty-Level Calibration.** To address **under-adaptivity**, the pruning strategy is further adapted to task difficulty, i.e., for easier tasks, a larger proportion of reasoning steps are removed, encouraging the model to condense its reasoning more aggressively (see "compression" on the right of Fig. 2). The difficulty of a task is estimated, based on the pass rate of each problem, during rollout. From the estimated difficulty level, each problem set is categorized among three difficulty levels: easy, medium, and hard, with a higher pass rate indicating easier problems and vice versa. Each category is assigned a compression rate to determine the degree of redundant steps to prune from the reasoning trajectory, with a higher compression for easier problems and a lower compression for hard problems. In addition, to keep these constraints adaptive to the amount of redundancy in the steps, we calculate the *uniformity* of the attention score distribution. When the distribution of $\{s_j\}$ is close to uniform, indicating that no step or token within a step stands out as significantly more important, the compression rate is reduced to avoid removing potentially useful reasoning steps. More details on calculating the uniformity score can be found in Appendix A.8.3. The difficulty estimate d is further incorporated with the reward calculation described below.

## 2.3 REWARDS

Following standard GRPO practice of having a verifiable reward system (Shao et al., 2024), our setup comprises three different reward signals to guide the model to generate correct adaptive length responses based on the difficulty of the task:

- **Correctness Reward (CR):** A high-weight reward is assigned to outputs that produce the correct final answer. A high score over other rewards is used to ensure that correctness remains the primary optimization objective, regardless of the reasoning trajectory length.

- **Format Reward:** A structure reward to ensure the presence of special delimiter tokens such as `<think>` and `</think>`, ensuring that trajectory $r$ and final answer $a$ are easily distinguishable.
- **Length Reward (LR):** To regulate the verbosity of the reasoning process, we define a length-based reward that penalizes unnecessarily long reasoning traces while adapting to task difficulty. Based on our initial experiment, simply favoring shorter rollouts led to a drastic decrease in response length along with model accuracy; therefore we introduce a sigmoid-based smoothing mechanism that provides a soft bonus ($\beta$) for rollouts beyond the median length. This prevents sharp drops in reward for slightly longer reasoning and helps stabilize training. During each training step, rollouts are partitioned into bins according to their calculated difficulty. As mentioned above, we use the pass rate of the rollouts to categorize them into three difficulty bins: easy, medium and hard. For each bin, we maintain a different distribution $\mathcal{L}_d = \{\ell_1, \ell_2, \ldots, \ell_m\}$ for each difficulty category, where each $\ell_i$ denotes the reasoning length of a rollout within that difficulty category d. Let $\ell$ be the length of the current rollout. The normalized length score is computed as: $L_{\text{norm}} = (L_{\text{max}} - \ell)/\max(L_{\text{max}} - L_{\text{min}}, \epsilon)$, where $\epsilon > 0$ prevents division by zero and $L_{\text{min}} = \min(\mathcal{L}), L_{\text{max}} = \max(\mathcal{L})$. To avoid a sharp cutoff around the median, we add a smooth bonus term:
$$\beta = 1/\left(1 + \exp\left(\frac{\ell - \text{median}(\mathcal{L})}{0.1 \times \text{median}(\mathcal{L})}\right)\right),$$
where $\text{median}(\mathcal{L}) = $ median of the set. The final length reward becomes $r_{\text{length}} = \max(L_{\text{norm}}, \beta)$. Note that length reward is only provided to a rollout if it reaches a final correct answer. Moreover, to ensure stability when calculating $L_{min}, L_{max}$, and medium($\mathcal{L}$), we maintain a sliding window over the last 10 steps for each difficulty bin, thereby avoiding drastic fluctuations during training.

The final reward for each rollout during GRPO training is the combination of correctness, format, and length rewards (c.f. range of each reward in Appendix A.10.2).

## 3 EXPERIMENTAL SETUP

**Models.** We adopt two reasoning models, DeepSeek-R1-Distill-Qwen-7B (DeepSeek-AI et al., 2025) (Deepseek-Qwen-7B) and Qwen3-4B (Team, 2025) as our base models.

**Datasets.** We train the model using DAPO-Math-17k (Yu et al., 2025), a math dataset that has verifiable answer. For evaluation, we use a diverse set of benchmarks, including AIME (AIME, 2024), AMC (AMC, 2023), GPQA-D (Rein et al., 2023), OverthinkingBench/ UnderthinkingBench (Aggarwal et al., 2025), and Big Bench Extra Hard (BBEH) (Kazemi et al., 2025). Among the evaluation datasets, only AIME and AMC are math-specific, while the remaining benchmarks represent out-of-distribution settings. Further dataset details and their sizes are provided in Appendix A.1.

**Evaluation.** For each evaluation run, we set temperature to 1.0, and the maximum response length is set to 10k. For each dataset, the mean accuracy and mean response length across 5 runs are reported. For the overthink split, we also report the $\text{AUC}_{\text{OAA}}$ (Aggarwal et al., 2025), directly used from their work. To aggregate performance across all thresholds, we compute the Area Under the Curve (AUC). More details about these metrics can be found in Appendix A.10.3. Intuitively, a higher $\text{AUC}_{\text{OAA}}$ indicates that the model sustains stronger accuracy while minimizing unnecessary reasoning across thresholds. Following evaluation from Aggarwal et al. (2025) for computing the OptimalThinkingBench score, we combined the $\text{AUC}_{\text{OAA}}$ from OverthinkingBench and accuracy from UnderthinkingBench into a single F1 score.

**Training.** During the GRPO rollout, we keep a high temperature of 1.0 and sample 8 rollouts at each step. Due to computational constraints, we set the maximum response length to 10k (see Appendix A.10.3 for other hyperparameter details). For difficulty calibration, we bin problems into easy, medium, and hard categories, assigning the categories decreasing compression scores.

**Baselines.** We compare TRAAC with 5 strong baselines: **(1) Base model:** off-the-shelf reasoning model, **(2) TokenSkip:** An SFT based baseline as described by Xia et al. (2025) that fine-tunes the model over compressed CoT training data. **(3) L1-Max**: An RL framework proposed by Aggarwal & Welleck (2025) that optimizes for accuracy while adhering to user-specific length constraints. We used the constraint "Think for a maximum of 10000 tokens." during its training. **(4) LC-R1:** A compression-based RL framework by Cheng et al. (2025) that uses an externally trained model to remove invalid portions of the thinking process. **(5) AdaptThink:** Different from the

Table 1: Performance comparison of TRAAC with various baselines. Each model is evaluate across various benchmarks, and Acc: accuracy(%) and Len: average Response Length(k) are reported. TRAAC on average shows the highest performance gain.

| Method | AIME | | AMC | | GPQA-D | | BBEH | | Average | |
|---|---|---|---|---|---|---|---|---|---|---|
| | Acc.↑ | Len.↓ | Acc.↑ | Len.↓ | Acc.↑ | Len.↓ | Acc.↑ | Len.↓ | Acc.↑ | Len.↓ |
| **Qwen3-4B** | | | | | | | | | | |
| Base Model | 27.64 | 9.2 | 68.19 | 7.0 | 45.18 | 7.6 | 18.28 | 6.7 | 39.8 | 7.6 |
| TokenSkip | 5.84 | 9.6 | 27.71 | 8.7 | 32.32 | 7.8 | 11.91 | 7.2 | 19.4 | 8.3 |
| L1-Max | 30.11 | 7.1 | 63.61 | 5.8 | 43.23 | 5.8 | 14.91 | 5.0 | 38.0 | 5.9 |
| LC-R1 | 13.48 | 2.6 | 56.38 | 1.7 | 26.67 | 1.5 | 12.35 | 1.9 | 27.2 | 1.9 |
| Adapt Think | 36.63 | 8.4 | 72.77 | 5.8 | 44.04 | 6.7 | 7.87 | 6.2 | 40.3 | 6.8 |
| TRAAC | **45.45** | 6.7 | **79.52** | 4.2 | **47.21** | 4.2 | **20.59** | 4.3 | **48.2** | 4.8 |
| **DeepSeek-R1-Distill-Qwen-7B** | | | | | | | | | | |
| Base Model | 33.71 | 8.2 | 74.22 | 5.7 | 43.55 | 7.1 | 10.61 | 5.9 | 40.5 | 6.7 |
| TokenSkip | 24.94 | 8.5 | 52.05 | 6.8 | 34.24 | 7.0 | 6.30 | 6.4 | 29.4 | 7.2 |
| L1-Max | 31.01 | 3.1 | 75.90 | 2.2 | 23.54 | 1.9 | **13.43** | 2.1 | 36.0 | 2.3 |
| LC-R1 | 6.07 | 4.0 | 37.35 | 3.5 | 28.78 | 2.5 | 9.09 | 1.7 | 20.3 | 2.9 |
| Adapt Think | **38.88** | 7.1 | 75.66 | 4.1 | 19.29 | 4.8 | 6.17 | 5.2 | 35.0 | 5.3 |
| TRAAC | 38.60 | 7.3 | **77.83** | 4.5 | **47.31** | 6.2 | 11.55 | 5.2 | **43.8** | 5.8 |

above baselines, AdaptThink is an adaptive RL framework described by Zhang et al. (2025b), that enables reasoning models to choose between "thinking" and "no-thinking" modes and poses it as a constraint optimization problem that encourages the model to choose no-thinking while maintaining performance. Prompts used for all baselines in Appendix A.10.5.

## 4 RESULT AND DISCUSSION

### 4.1 MAIN RESULTS

**TRAAC improve both performance and efficiency.** Tables 1 show the performance of TRAAC compared to other baselines on AIME, AMC, GPQA-D, BBEH (Big Bench Extra Hard) benchmarks. TRAAC (Qwen3-4B) achieves an average accuracy improvement of 8.4% while reducing reasoning length by 36.8% compared to the base model. Similarly, TRAAC (Deepseek-Qwen-7B) improves accuracy by 3.3% with a 13.4% reduction in length. When compared to the SFT baseline Token-Skip (Xia et al., 2025), TRAAC outperforms in terms of performance and efficiency for both models, Qwen3-4B and Deepseek-Qwen-7B. Similarly, L1-Max (Aggarwal & Welleck, 2025), an RL-based method that penalizes long responses, also solely focuses on efficiency gains, at a slight cost of overall performance. Additionally, the compression-based RL framework LC-R1 (Cheng et al., 2025) improves the efficiency of the model at the cost of a 12.6% drop for Qwen3-4B and 20.2% drop for Deepseek-Qwen-7B, when compared with base models, respectively. On average for Qwen3-4B, TRAAC outperforms L1-Max by 10.2% on Qwen3-4B and by 7.9% on Deepseek-Qwen-7B. Similarly, TRAAC also outperforms LC-R1 by 21% on Qwen3-4B and 23% on Deepseek-Qwen-7B. Moreover, given the same token budget, of approximately 7k, TRAAC (Qwen3-4B) on AIME outperforms L1-Max by 15%. These results highlight that, unlike methods that prioritize only efficiency, TRAAC simultaneously delivers both higher accuracy and shorter reasoning traces.

**TRAAC generalizes across domains.** Recall that for training TRAAC we used data from DAPO-Math-17k (Yu et al., 2025), which is a math reasoning dataset. In addition to math datasets, we also evaluate TRAAC on several out-of-domain (OOD) tasks, including GPQA-D, BBEH, OverthinkingBench, and UnderthinkingBench (Table 2). Among these OOD tasks, TRAAC shows an average improvement of 3% on Qwen3-4B and 2.8% on Deepseek-Qwen-7B compared to the base model, with improvement as high as 6.8% on UnderthinkingBench, which covers 100 diverse reasoning tasks from Reasoning Gym (Stojanovski et al., 2025). In addition, TRAAC reduces reasoning tokens by 40% on Qwen3-4B and 20% on Deepseek-Qwen-7B, demonstrating substantially higher efficiency while also boosting accuracy across benchmarks. This indicates that TRAAC learns a generalizable compression strategy that transfers from math to other reasoning domains.

Table 2: Performance of TRAAC and various baselines on OptimalThinkingBench (OTB). For Un-derthinkingBench we report the Acc: Accuracy(%), and Len: Average Response length(k). For OverthinkingBench, in addition to Acc. and Len. we also report the $AUC_{OAA}$.

| Method | OverthinkingBench | | | UnderthinkingBench | | OTB |
|---|---|---|---|---|---|---|
| | Acc.↑ | Len.↓ | $AUC_{OAA}$ ↑ | Acc.↑ | Len.↓ | F1↑ |
| **Qwen3-4B** | | | | | | |
| Base Model | **90.02** | 1.2 | 80.06 | 34.33 | 7.1 | 48.05 |
| TokenSkip | 78.15 | 3.5 | 57.88 | 14.80 | 7.9 | 23.57 |
| L1-Max | 87.22 | 0.9 | 1.11 | 21.27 | 6.3 | 2.10 |
| LC-R1 | 78.62 | 0.3 | 64.20 | 14.95 | 1.3 | 24.25 |
| Adapt Think | 68.83 | 8.2 | 63.44 | 18.80 | 6.0 | 29.01 |
| TRAAC | 89.79 | 0.6 | **85.06** | **41.09** | 4.7 | **55.41** |
| **DeepSeek-R1-Distill-Qwen-7B** | | | | | | |
| Base Model | 78.45 | 0.9 | 72.38 | 12.69 | 6.2 | 21.60 |
| TokenSkip | 57.03 | 3.9 | 40.77 | 8.55 | 7.2 | 14.13 |
| L1-Max | 73.18 | 1.0 | 66.01 | 20.07 | 2.0 | 30.78 |
| LC-R1 | 76.08 | 0.9 | 69.81 | 7.16 | 2.5 | 12.99 |
| Adapt Think | 73.41 | 0.4 | 70.72 | 13.13 | 4.6 | 22.14 |
| TRAAC | **81.81** | 1.0 | **72.89** | **22.30** | 5.9 | **34.15** |

**TRAAC learns to adaptively allocate token budget.** Among the baselines in Tables 1 and 2, we also compare TRAAC against an adaptive RL method, AdaptThink (Zhang et al., 2025b), which teaches the model to use distinct "thinking" vs. "non-thinking" modes for hard and easy prob-lems, respectively. On Qwen3-4B, TRAAC outperforms AdaptThink by 7.9% while also reducing tokens by 29.4%, highlighting that a flexible adaptive strategy is more effective in handling diverse problem difficulties. Table 2 further tests on the OverthinkingBench/UnderthinkingBench (Aggar-wal et al., 2025). OverthinkingBench is designed to measure excessive use of thinking tokens on simple queries. On the other hand, UnderthinkingBench evaluates how necessary "thinking" is based on problem difficulty. Taken together, TRAAC improves overall F1 performance by 7.36% on Qwen3-4B, and 12.55% on Deepseek-Qwen-7B over base model, indicating that TRAAC enables the model to avoid both overthinking on simple problems and underthinking on complex ones(Aggarwal et al., 2025). Against AdaptThink, TRAAC achieves a 26% gain on Qwen3-4B and a 12% gain on Deepseek-Qwen-7B, underscoring its ability to adaptively allocate reasoning effort and adjust token budgets based on problem difficulty. On OverthinkingBench, we measure overthinking using the AUCOAA metric, which rewards models that solve very easy problems correctly while using min-imal tokens (ideally 0). Compared to the base model, TRAAC (Qwen3-4B) improves AUCOAA by 5% and Deepseek-Qwen-7B by 0.5%. Relative to AdaptThink, TRAAC gains 21.6% for Qwen3-4B and 6.9% for Deepseek-Qwen-7B.

## 4.2 ABLATIONS AND ANALYSIS

To understand the importance of each component of the training setup we conducted an ablation study, removing each component of our method. Table 3 and Table 4 show the performance of these ablations compared with the base model. Specifically, we start with the base model and the ablations: **(i) Base Model + CR:** The base model trained with GRPO using only the correctness reward, **(ii) Base model + CR + LR:** The base model trained with GRPO using both correctness and length rewards, but without difficulty-level calibration, **(iii) Base model + CR + LR + Compression:** The base model trained with GRPO using correctness and length rewards, along with the compression module, with no difficulty-level calibration. Our findings are as follows.

**Combining difficulty-adaptiveness and attention-based compression is crucial for accuracy and efficiency.** Table 3 shows that on Qwen3-4B, removing the difficulty-based calibration (Base Model + CR + LR + compression) reduces the average performance across AIME, AMC, GPQA-D, and BBEH by 3.4%, while also making the model less efficient by 23.8%. Additionally, removing the attention-based compression (Base Model + LR + CR) leads to a further drop in performance by 0.3%. Similarly, on OptimalThinkingBench (Table 4), we observe a comparable degradation: the

Table 3: Ablation Results of TRAAC on Qwen3-4B and Deepseek-Qwen-7B tested across 4 datasets: AIME, AMC, GPQA-D, and BBEH. Each component addition adds to the previous method.

| Method | AIME | | AMC | | GPQA-D | | BBEH | | Average | |
|---|---|---|---|---|---|---|---|---|---|---|
| | Acc.↑ | Len.↓ | Acc.↑ | Len.↓ | Acc.↑ | Len.↓ | Acc.↑ | Len.↓ | Acc.↑ | Len.↓ |
| **Qwen3-4B** | | | | | | | | | | |
| Base Model | 27.64 | 9.2 | 68.19 | 7.0 | 45.18 | 7.6 | 18.28 | 6.7 | 39.8 | 7.6 |
| + CR | 44.36 | 7.9 | 77.35 | 5.5 | 46.29 | 5.7 | 18.13 | 5.2 | 46.5 | 6.1 |
| + LR | 37.84 | 4.5 | 77.35 | 2.4 | 44.06 | 2.3 | 18.57 | 2.1 | 44.5 | 2.8 |
| + Compression | 38.37 | 8.1 | 75.90 | 5.5 | 46.40 | 6.2 | 18.41 | 5.4 | 44.8 | 6.3 |
| TRAAC | **45.45** | 6.7 | **79.52** | 4.2 | **47.21** | 4.2 | **20.59** | 4.3 | **48.2** | 4.8 |

Table 4: Ablation Results of TRAAC (Qwen3-4B and Deepseek-Qwen-7B) on OptimalThinking-Bench (OTB). Each component addition adds to the previous method.

| Method | OverthinkingBench | | | UnderthinkingBench | | OTB |
|---|---|---|---|---|---|---|
| | Acc.↑ | Len.↓ | $AUC_{OAA}$ ↑ | Acc.↑ | Len.↓ | F1↑ |
| **Qwen3-4B** | | | | | | |
| Base Model | 90.02 | 1.2 | 80.06 | 34.33 | 7.1 | 48.1 |
| + CR | 90.02 | 0.9 | 78.86 | 37.06 | 5.7 | 50.4 |
| + LR | **90.94** | 0.4 | 75.86 | 29.62 | 2.3 | 42.6 |
| + Compression | 90.12 | 0.9 | 80.41 | 36.51 | 6.0 | 50.2 |
| TRAAC | 89.79 | 0.6 | **85.06** | **41.09** | 4.7 | **55.4** |

F1 score decreases by 5.2% when task-difficulty level calibration is removed and drops further by 7.6% when the attention-based compression module is also removed. These results highlight that a combination of task-difficulty calibration and attention-based compression is crucial for achieving both high performance and efficiency gains across tasks.

**TRAAC adapts to task difficulty.** To further understand the level of adaptivity of TRAAC compared to other methods, we plot the relative compression ratio and absolute accuracy gains (w.r.t. the base model) in Fig. 3 as a function of task difficulty. Here, we rank tasks in order of increasing difficulty. We conduct these experiments on SuperGPQA (Team et al., 2025) – a benchmark to evaluate model knowledge and reasoning capabilities, which is stratified into easy, medium, and hard splits, and BBH (Big Bench Hard) (Suzgun et al., 2022) – an easier version of BBEH. To get oracle difficulty ratings, we rank the datasets by the performance of frontier models on them (Kazemi et al., 2025; Team et al., 2025), with harder datasets being those with lower performance. From Fig. 3(a), we see that as the difficulty of the dataset increases from left to right, the compression rate steadily drops for TRAAC, underscoring its ability to compress more for easier tasks and less for difficult tasks. However, without task-difficulty level calibration, the compression rate remains roughly uniform across the tasks. Fig. 3(b) highlights the performance difference, and shows that even with more compression, TRAAC always maintains higher accuracy than Qwen3-4B + CR + LR + compression, reiterating the effectiveness of adapting to problem difficulty in TRAAC. Moreover, most of the accuracy gains stems from harder problems, indicating the average accuracy gains seen in Table 1 come from difficulty-adaptive thinking. Deepseek-Qwen-7B results are shown in Appendix A.7 and follow a similar trend as Qwen3-4B.

**TRAAC scales to larger response length, maintaining its improvement.** During TRAAC training, we set a maximum token budget of 10k. To test the scalability of our method, we increase the max training and test-time response length to 15k. Table 5 shows the accuracy and average response length for AIME, AMC, and GPQA-D datasets, for the Qwen3-4B and TRAAC with increased token budget. Similar to the

Table 5: TRAAC with 15k training and test-time response length. For each dataset, Accuracy (%) and Response Length (in ×1000 tokens) are reported.

| | AIME | AMC | GPQA-D |
|---|---|---|---|
| Qwen3-4B | 47.74 / 12.3 | 77.11 / 8.5 | 49.64 / 8.6 |
| TRAAC | **51.93** / 9.7 | **81.68** / 6.6 | **51.27** / 6.2 |

prior results, we see an average accuracy improvement of 3.5% and 23.4% efficiency gains. This underscores that scaling TRAAC still shows consistent gains for both accuracy and efficiency.

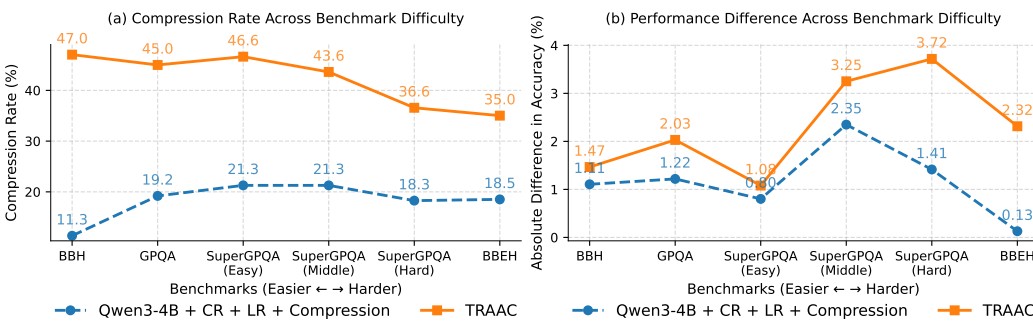

Figure 3: (a) Relative change in compression rate of TRAAC and Qwen3-4B + Compression compared to Qwen3-4B across varying problem difficulty. (b) Absolute accuracy drop of TRAAC and Qwen3-4B + Compression compared to Qwen3-4B across varying problem difficulty.

**Attention-based compression identifies redundant steps effectively.** To help understand the efficiency of the adaptive, attentive compression module, we replace the attention-based compression with random step compression or confidence-based compression. At each training step, instead of using attention as a metric, reasoning steps are pruned either randomly

Table 6: Ablation on Qwen3-4B: comparing TRAAC with pruning random and least confident steps. For each dataset, Accuracy(%) / Response length (k) is reported.

| Pruning Strategy | AIME | AMC | GPQA-D |
|---|---|---|---|
| Random Steps | 29.54 / 6.5 | 66.74 / 4.1 | 42.94 / 3.2 |
| Least Confidence | 32.35 / 5.8 | 71.08 / 3.4 | 47 / 3.0 |
| TRAAC | **45.45** / 6.7 | **79.52** / 4.2 | **47.2** / 4.2 |

or steps with the least confidence (complete details on how confidence is calculated are in Appendix A.5). Table 6 compares TRAAC (Qwen3-4B) with random steps and least confidence. Relative to TRAAC, random step pruning shows an average of 11% accuracy drop, and similarly, pruning the least confidence steps leads to a 7.25% accuracy drop. This highlights the efficacy of using attention-based compression in TRAAC.

**TRAAC reward design achieves the highest performance.** During TRAAC training, we use a correctness reward of $+4/0$, a format reward of $+1/0$, and a length reward of $+2/0$. The length reward is kept *adaptive*: rollouts receive a positive length reward only if the final answer is correct; otherwise, all length rewards are set to zero. This ensures that correctness is always prioritized over efficiency,

Table 7: Reward ablation results comparing different correctness and length reward configurations. For each dataset, Accuracy (%) and Response Length (in $\times 1000$ tokens) are reported.

| | AIME | AMC | GPQA-D |
|---|---|---|---|
| TRAAC$_{\text{reduced correctness}}$ | 29.96 / 6.2 | 71.32 / 3.9 | 47.7 / 3.6 |
| TRAAC$_{\text{non adaptive}}$ | 5.33 / 0.6 | 34.87 / 0.7 | 29.79 / 0.5 |
| TRAAC | 45.45 / 6.7 | 79.52 / 4.2 | 47.21 / 4.2 |

which is reflected in the larger magnitude of the correctness reward. To study the impact of this reward design, we conduct two additional ablations: (i) **Reduced correctness reward:** correctness reward lowered to $+1$, with length reward still adaptive to correctness. (ii) **Non-adaptive length reward:** correctness reward lowered to $+1$, and length reward made independent of final-answer correctness. Table 7 compares these variants with the full TRAAC setup across AIME, AMC, and GPQA-D. The ablation results show that reducing the correctness reward maintains reasonable performance on AMC and GPQA-D but causes a substantial accuracy drop on AIME. Removing the dependency of the length reward on correctness leads the model to exploit the reward by minimizing output length, resulting in severe performance degradation across all datasets.

## 5 RELATED WORK

In the past years, reasoning performance of language models has vastly improved via the introduction of chain-of-thoughts (Wei et al., 2023), parallel scaling through self-consistency (Wang et al., 2023), and best-of-$N$ sampling (Lightman et al., 2023). More recently, several works have found sequential scaling – i.e., increasing the number of reasoning tokens – to be the most effective ap-

proach (Muennighoff et al., 2025), especially when combined with online reinforcement learning or distillation from such models (Aggarwal & Welleck, 2025; Shao et al., 2024; DeepSeek-AI et al., 2025). Consequently, the area of efficient reasoning – maintaining high performance from sequential scaling with minimal token usage – has become a central research focus (Chen et al., 2024; Marjanović et al., 2025; Wu et al., 2025). To this end, prior works compress or prune chain-of-thoughts via early exiting (Zhang et al., 2025a; Fu et al., 2025), train models under pre-specified budgets (Aggarwal & Welleck, 2025), learn thoughts latently without generating them (Hao et al., 2025), use supervised finetuning to avoid overthinking (Xia et al., 2025; Cheng et al., 2025; Lu et al., 2025), or add length-based penalties for conciseness (Arora & Zanette, 2025; Hou et al., 2025). However, this line of work does not *explicitly* account for varying problem difficulty, instead relying on the model to learn to allocate budget implicitly; in contrast, TRAAC introduces difficulty-based supervision for budget allocation. Moreover, prior approaches typically address only overthinking – reducing output length at the cost of performance drops – whereas we tackle both over- and underthinking.

Improving *both* reasoning performance and efficiency requires a more *adaptive* approach through explicit training. Prior work such as Zhang et al. (2025b) frames adaptivity as a binary decision of *whether* to think, whereas we argue that for harder problems it must involve deciding *how much* to think – and empirically outperform this baseline in Appendix 4.1. A similar insight appears in planning, where Saha et al. (2025) show that mixing "system 1" and "system 2" reasoning within the same instance outperforms a binary choice between them. Shen et al. (2025) pursue difficulty-adaptive training via repeated sampling and offline preference optimization to prefer shorter responses. In contrast, TRAAC provides attention-based supervision in the compression module through online RL (DeepSeek-AI et al., 2025). Unlike concurrent work by Choi et al. (2025), who prune redundant tokens post hoc, our method adapts compression during training itself – yielding difficulty-aware reasoning and improved test-time efficiency without generating unnecessary tokens.

## 6 CONCLUSION

We introduced TRAAC, a post-training RL method that operates online and uses a difficulty-adaptive, attention-based compression module. Through its adaptive attentive compression, TRAAC is able to prune its reasoning steps adaptively based on the task difficulty. TRAAC addresses the issue of under-adaptivity, which helps improve both performance and efficiency, as thinking longer on harder problems helps in better exploration, and thinking shorter on easier problems avoids wasting of test-time compute. Moreover, our method also shows strong generalizability, with evaluation done on various OOD tasks. Through our analysis and ablation, we further verify that our adaptive method can provide fine-grained adjustments to the thinking budget based on the difficulty of the problem, and a combination of task-difficulty calibration and attention-based compression helped achieve both accuracy and efficiency gains.

## ETHICS STATEMENT

TRAAC is a reinforcement learning method that rewards models based on the correctness of the final answer. Therefore, the trained LLMs may still generate hallucinations, since their intermediate reasoning steps are neither guided nor evaluated – only the final result is checked. This means outputs from TRAAC can pose risks of misinformation or hallucination. Future work is needed to more thoroughly evaluate and mitigate these issues.

## REPRODUCIBILITY STATEMENT

We are making our code available in the supplementary materials to help reproduce our findings. We also provide detailed descriptions, hyperparameters, and prompts about the implementation of TRAAC in Appendix A.10.

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

# A  APPENDIX

## A.1  DATASET DETAILS

We evaluated the model on various benchmarks:

- AMC: All questions come from AMC12 2022, AMC12 2023, and have been extracted from the AOPS wiki page. Total Count: 83
- AIME: All questions come from AIME 22, AIME 23, and AIME 24, and have been extracted directly from the AOPS wiki page. Total Count: 90
- GPQA-D: It is a multiple-choice dataset covering physics, biology, and chemistry. Total Count: 198
- BBEH: A benchmark designed to push the boundaries of LLM reasoning evaluation. BBEH replaces each task in BBH with a novel task that probes a similar reasoning capability but exhibits significantly increased difficulty. Total Count: 460
- OptimalThinkingBench: A unified benchmark that jointly evaluates overthinking and under-thinking in LLMs and also encourages the development of optimally-thinking models that balance performance and efficiency. Two sub benchmarks: OverthinkingBench, featuring simple queries in 72 domains, and UnderthinkingBench, containing 11 challenging reasoning tasks. UnderthinkingBench count: 550, OverthinkingBench count: 607.
- BBH: a suite of 23 challenging BIG-Bench tasks. Total Count: 2115
- SuperGPQA: A comprehensive benchmark designed to evaluate the knowledge and reasoning abilities of Large Language Models (LLMs) across 285 graduate-level disciplines. Each problem is also categorized as easy, medium and hard. 540 problems for each difficulty category, so the total count is 1620.

To calculate the accuracy, we adopt Math-Verify [2]. For UnderthinkingBench accuracy calculation, we used the evaluation scripts from Reasoning-Gym (Stojanovski et al., 2025)

## A.2  COMPUTATIONAL COST ANALYSIS OF TRAINING TRAAC

To understand the computational cost of training TRAAC vs other RL-based methods (L1-max, Adapt-Think), we compare TRAAC with RL baselines on training time and FLOPs.

### A.2.1  TRAINING TIME

The GRPO algorithm mainly consists of three stages: (i) **Rollout**: the LLM produces multiple responses for a given prompt; (ii) **Scoring**: a scalar reward is assigned to each response; (iii) **Policy optimisation**: the LLM is updated by optimising the total objective. Since we use the `math-verify` library – a rule-based expression system that does not require additional LLM calls for reward computation – the cost of scoring is negligible.

Table 8 reports the breakdown of training time for TRAAC compared to RL baselines. For each method, we show the wall-clock time (in seconds) for the first training step, split into rollout time, policy optimisation time, and total time.

- **Base Model + CR**: GRPO with correctness reward only.
- **Base Model + CR + LR**: GRPO with correctness and length rewards (no difficulty calibration).

Other RL baselines such as L1-MAX and ADAPTTHINK are also variants of Base Model + CR with an additional length reward term.

---

[2] Huggingface Math-Verify

Table 8: Training time breakdown for TRAAC and RL baselines during the first GRPO training step.

| Method | Rollout (sec) | Optimise Policy (sec) | Total Time (sec) | Hardware |
|---|---|---|---|---|
| Base Model + CR | 250 | 87.5 | 397.5 | H100 |
| Base Model + CR + LR | 222 | 88 | 375 | H100 |
| TRAAC | 418 | 88 | 583 | H100 |

### A.2.2 FLOPs

The majority of the difference between TRAAC and other RL-based methods lies in the rollout strategy used. TRAAC rollout consists of three stages: (i) **Generation**: producing the initial reasoning steps; (ii) **Attention-based compression**: computing attention scores for each reasoning step and compressing the trajectory; (iii) **Answer generation**: generating the final answer based on the compressed chain of thought. Table 9 compares the FLOPs required to generate 20 training examples.

Table 9: FLOPs comparison for generating 20 training examples using different rollout strategies.

| Method | FLOPs Used |
|---|---|
| Base Model + CR | $1.65 \times 10^{15}$ FLOPs |
| TRAAC | $3.84 \times 10^{15}$ FLOPs |

Most RL baselines only perform the initial generation step. In contrast, TRAAC adds an additional attention-computation stage, yet keeps the overall FLOPs in the same order of magnitude—while producing higher-quality reasoning trajectories. Even though TRAAC incurs an increase in training time and FLOPs for the initial batches, mainly due to its multi-stage generation, the overhead remains modest. Moreover, during inference, TRAAC makes the model more efficient, effectively reducing the computational cost at test time.

### A.2.3 COST AMORTIZES AS TRAINING PROGRESSES

The additional overhead introduced at the beginning of the training quickly amortises as the training progresses. As the model learns to shorten its generated reasoning traces, its computational cost – including both FLOPs and time per step steadily decreases. To show this empirically, Figure 4 shows the time-per-step curve for training DeepSeek-Qwen-7B. As illustrated in the figure, TRAAC begins with a higher step time compared to the Base Model + CR baseline, but the gap closes rapidly. Around mid-training, the two curves match closely, and in later steps, TRAAC consistently becomes more efficient – ultimately achieving a lower step time than the baseline. This confirms that while TRAAC introduces an initial overhead, its adaptive reduction of reasoning length leads to substantial efficiency gains, resulting in lower computation over the course of training.

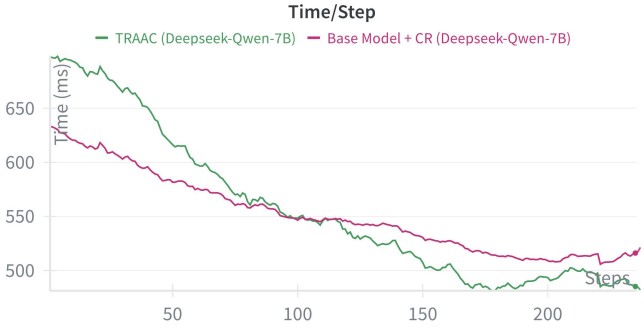

Figure 4: Time per step across training (Deepseek-Qwen-7B)

To highlight the advantage gained from training TRAAC in terms of FLOPs we calculated the average FLOPs required during inference. We calculated the total amount of FLOPs for generating 80 examples from AMC dataset. The table 10 show the total amount of FLOPs required for both TRAAC and Base Model + CR.

Table 10: Inference compute comparison for TRAAC vs. Base Model + CR on 80 AMC questions.

| Method | Total FLOPs (80 questions) | Average FLOPs per question |
|--------|----------------------------|----------------------------|
| Base Model + CR (Qwen3-4B) | $3.7 \times 10^{15}$ FLOPs | $4.6 \times 10^{13}$ FLOPs |
| TRAAC (Qwen3-4B) | $2.7 \times 10^{15}$ FLOPs | $3.3 \times 10^{13}$ FLOPs |

TRAAC yields a substantial reduction in inference compute. As shown above, TRAAC requires $2.7 \times 10^{15}$ FLOPs to answer 80 AMC questions, compared to $3.7 \times 10^{15}$ FLOPs for the *Base Model + CR*. This corresponds to a $27.3\%$ reduction in inference FLOPs, or a savings of $1.2664 \times 10^{13}$ FLOPs per question. Although TRAAC incurs additional compute at the beginning of training, this overhead amortizes quickly as training progresses. This cost is balanced rapidly at inference time.

Therefore, with the above two experiments, we show that TRAAC not only amortizes quickly during inference but also makes training more efficient within the first 100 steps.

### A.3 AGENTIC EVALUATION OF TRAAC

Including diverse test benchmarks allows us to robustly assess the out-of-distribution generalization capability of TRAAC. In addition, we conduct an evaluation on the agentic, multi-turn benchmark **MINT** (Wang et al., 2024), which measures an LLM's ability to solve complex tasks through multi-step interactions and tool use. In MINT, LLMs are tasked with solving problems under different interaction limits $k \in \{1, 2, 3, 4, 5\}$, without natural-language feedback. Performance is measured through: (1) the absolute success rate (SR), and (2) the average response length. Table 11 compares the base model (Qwen3-4B) and TRAAC across these metrics for three task categories: code generation, decision making, and reasoning.

Table 11: MINT benchmark results for Base Model (Qwen3-4B) and TRAAC across interaction limits $k \in \{1, 2, 3, 4, 5\}$. Metrics include success rate (SR, %) and average response length.

| Task | Method | $k = 1$ | | $k = 2$ | | $k = 3$ | | $k = 4$ | | $k = 5$ | |
|------|--------|---------|-----|---------|-----|---------|-----|---------|-----|---------|-----|
| | | SR | Len | SR | Len | SR | Len | SR | Len | SR | Len |
| code_generation | Base Model | 0.74 | 0.5 | 58.09 | 3.7 | 58.09 | 4.8 | 59.56 | 5.7 | 59.56 | 7.1 |
| | TRAAC | 49.26 | 1.7 | 58.82 | 2.6 | 56.62 | 3.3 | 59.56 | 3.5 | 58.82 | 3.8 |
| decision_making | Base Model | 0.00 | 0.5 | 11.19 | 2.4 | 17.16 | 2.5 | 30.60 | 2.5 | 33.58 | 3.0 |
| | TRAAC | 0.00 | 0.5 | 8.21 | 1.0 | 21.64 | 1.2 | 35.07 | 1.3 | 40.30 | 1.6 |
| reasoning | Base Model | 19.94 | 0.5 | 76.58 | 1.9 | 80.38 | 2.2 | 79.75 | 2.5 | 79.75 | 2.4 |
| | TRAAC | 66.46 | 0.8 | 76.90 | 1.1 | 81.65 | 1.2 | 79.11 | 1.3 | 81.96 | 1.2 |
| avg_micro | Base Model | 10.92 | 0.5 | 57.34 | 2.4 | 60.75 | 2.9 | 63.82 | 3.2 | 64.51 | 3.6 |
| | TRAAC | 47.27 | 1.0 | 57.00 | 1.4 | 62.12 | 1.7 | 64.51 | 1.8 | 67.06 | 1.9 |

When examining the average micro-aggregated performance across all interaction limits, TRAAC consistently matches or exceeds the base model. TRAAC improves the average success rate by $8.12\%$ while simultaneously reducing response length by $38.3\%$. This demonstrates that TRAAC not only strengthens performance on agentic, multi-turn tasks but also makes the model substantially more efficient in its interactions.

### A.4 WHY CHOOSE RL IN TRAAC

Choosing reinforcement learning (RL) rather than supervised fine-tuning (SFT) to teach adaptive compression is motivated by two key reasons:

- **Generalization advantages of RL.** Prior work has repeatedly shown that RL-based methods yield significantly stronger generalization compared to SFT (Chu et al., 2025). In our own exper-

iments, the SFT-based baseline TokenSkip performs substantially worse than TRAAC, demonstrating that simply applying SFT on compressed outputs is insufficient.

- **Adaptive compression requires an online difficulty signal.** TRAAC relies on dynamically compressing reasoning trajectories according to the difficulty of each problem, where the difficulty signal itself is tightly coupled to the model's evolving capabilities. Because TRAAC learns this adaptivity during training, an online RL setting naturally allows difficulty estimates to improve alongside the model, enabling progressively better compression decisions.

SFT ON ATTENTION-BASED COMPRESSED TRAJECTORIES

For a direct comparison, we also train an SFT model using reasoning trajectories compressed via TRAAC's attention-based rollout. To generate the dataset, we use a larger model (Qwen3-32B) to produce 1.4k compressed rollouts, and then train a smaller model (Qwen3-4B) on this data. The table below reports the performance of the SFT model relative to the base model and TRAAC across three benchmarks: AIME, AMC, and GPQA. The SFT model matches the base model's accuracy while offering moderate efficiency gains. In contrast, TRAAC improves both accuracy and efficiency across all benchmarks.

Table 12: Comparison of TokenSkip, Base Model, SFT on attention-compressed rollouts, and TRAAC across AIME, AMC, and GPQA. Metrics include accuracy (%) and average response length.

| Method | AIME | | AMC | | GPQA | |
|---|---|---|---|---|---|---|
| | Acc. | Len. | Acc. | Len. | Acc. | Len. |
| TokenSkip | 5.84% | 9.6k | 27.71% | 8.7k | 32.32% | 7.8k |
| Base Model | 27.64% | 9.2k | 68.19% | 7.0k | 45.18% | 7.6k |
| SFT | 26.06% | 8.8k | 59.51% | 6.6k | 42.00% | 6.9k |
| TRAAC | **45.45%** | **6.7k** | **79.52%** | **4.2k** | **47.21%** | **4.2k** |

## A.5 CONFIDENCE BASED COMPRESSION

Similar to attention compression, where a score is calculated for each reasoning token, the confidence of the model is used to calculate the score, and based on the lowest average score, reasoning steps are removed. Algorithm 1 shows the pseudocode used to calculate the confidence of each token.

---

**Algorithm 1:** Token Confidence Calculation

**Input:** Top-$k$ token log-probabilities $L = \{\ell_1, \ell_2, \ldots, \ell_k\}$
**Output:** Confidence score $C$
**begin**
    // Convert log-probabilities to probabilities
    $p_j \leftarrow \exp(\ell_j)$ for each $\ell_j \in L$ ;
    // Normalize probabilities
    $Z \leftarrow \sum_{j=1}^{k} p_j$ ;
    $p_j \leftarrow p_j/Z$ for each $j$ ;
    // Compute entropy of distribution
    $H \leftarrow -\sum_{j=1}^{k} p_j \cdot \log(p_j + \epsilon)$ ;
    // Maximum entropy with $k$ tokens
    $H_{\max} \leftarrow \log(k)$ ;
    // Confidence is normalized inverse entropy
    $C \leftarrow 1 - (H/H_{\max})$ ;
    **return** $C$

---

## A.6 ATTENTION SCORE OVER FULL ATTENTION SCORE

During generation from a reasoning model, the `</think>` token marks the end of the reasoning process, and the answer tokens generated after `</think>` contain key summarized conclusions. Prior work (Choi et al., 2025) shows that `</think>` strongly attends to the critical reasoning steps needed to derive the final answer. To verify that attention score from `</think>` is more effective in practice, we ran an additional ablation that computes attention over the full context, including all reasoning tokens as well as the answer tokens generated after `</think>`. Using these attention scores, low-scoring steps were removed from the reasoning trajectory. The table below compares performance between using attention over the full rollout and using TRAAC.

| Method | AIME | | AMC | | GPQA | |
|---|---|---|---|---|---|---|
| | Acc. | Len. | Acc. | Len. | Acc. | Len. |
| TRAAC (full attention rollout) | 2.309% | 1.9k | 18.53% | 2.1k | 25.81% | 4.2k |
| TRAAC | 45.45% | 6.7k | 79.52% | 4.2k | 47.21% | 4.2k |

Table 13: Comparison of TRAAC with full-rollout attention pruning vs. standard TRAAC.

Accuracy drops sharply across all three datasets when attention is computed over the full rollout. This demonstrates that without relying on the `</think>` token for attention scoring, the model cannot reliably identify and prune redundant reasoning steps. Especially on AIME and AMC, we observe a substantial drop in efficiency, indicating that when attention is computed over the complete rollout – including both reasoning and final answers – the model struggles to determine which steps are informative versus unnecessary.

## A.7 DEEPSEEK ABLATION AND ANALYSIS

Table 14 and Table 15 present the ablation results for **(i) Base Model + CR:** The base model trained with GRPO using only the correctness reward, **(ii) Base model + CR + LR:** The base model trained with GRPO using both correctness and length rewards, but without difficulty-level calibration.

Table 14: Ablation Results of TRAAC on Qwen3-4B and Deepseek-Qwen-7B tested across 4 datasets: AIME, AMC, GPQA-D, and BBEH. Each component addition adds to the previous method.

| Method | AIME | | AMC | | GPQA-D | | BBEH | | Average | |
|---|---|---|---|---|---|---|---|---|---|---|
| | Acc.↑ | Len.↓ | Acc.↑ | Len.↓ | Acc.↑ | Len.↓ | Acc.↑ | Len.↓ | Acc.↑ | Len.↓ |
| | | | | **DeepSeek-R1-Distill-Qwen-7B** | | | | | | |
| Base Model | 33.71 | 8.2 | 74.22 | 5.7 | 43.55 | 7.1 | 10.61 | 5.9 | 40.5 | 6.7 |
| + CR | 35.81 | 7.6 | 78.55 | 4.9 | 45.99 | 6.1 | **11.74** | 5.1 | 43.0 | 5.9 |
| + LR | 32.73 | 6.0 | **79.04** | 3.3 | 45.99 | 3.5 | 11.51 | 2.7 | 42.3 | 3.9 |
| TRAAC | **38.60** | 7.3 | 77.83 | 4.5 | **47.31** | 6.2 | 11.55 | 5.2 | **43.8** | 5.8 |

Table 15: Ablation Results of TRAAC (Qwen3-4B and Deepseek-Qwen-7B) on OptimalThinking-Bench (OTB). Each component addition adds to the previous method.

| Method | OverthinkingBench | | | UnderthinkingBench | | OTB |
|---|---|---|---|---|---|---|
| | Acc.↑ | Len.↓ | AUC$_{OAA}$ ↑ | Acc.↑ | Len.↓ | F1↑ |
| | | | **DeepSeek-R1-Distill-Qwen-7B** | | | |
| Base Model | 78.45 | 0.9 | 72.38 | 12.69 | 6.2 | 21.6 |
| + CR | 79.51 | 0.8 | **73.36** | 17.05 | 5.7 | 27.7 |
| + LR | 78.06 | 0.4 | 72.61 | 14.69 | 3.0 | 24.4 |
| TRAAC | **81.81** | 1.0 | 72.89 | **22.30** | 5.9 | **34.1** |

## A.8 COMPRESSION MODULE

### A.8.1 PROMPT

For every reasoning trajectory, auxiliary prompt was appended at the end of the trajectory. The prompt is: "Time is up. I should stop thinking and now write a summary containing all key steps required to solve the problem.".

### A.8.2 SPECIAL TOKENS TO SPLIT TRAJECTORY TO CHUNKS

Below is the list that is used to split each reasoning trajectory into multiple reasoning steps.

```
split_tokens = [
    "Wait", "Alternatively", "Another angle", "Another approach", "But wait",
    "Hold on", "Hmm", "Maybe", "Looking back", "Okay", "Let me", "First",
    "Then", "Alright", "Compute", "Correct", "Good", "Got it",
    "I don't see any errors", "I think", "Let me double-check", "Let's see",
    "Now", "Remember", "Seems solid", "Similarly", "So", "Starting",
    "That's correct", "That seems right", "Therefore", "Thus"
]
```

### A.8.3 UNIFORMITY SCORE

Algorithm 2 presents the pseudocode for calculating the uniformity score, based on which the final compression rate is calculated.

---

**Algorithm 2:** Calculating Eviction Percentage Based on Attention Uniformity

---

**Input:** Step importance scores $\{s_1, s_2, \ldots, s_n\}$, target reduction $\tau$ (default: 0.25)
**Output:** Eviction percentage $e \in [0, 1]$
**Function** CALCULATEUNIFORMITYSCORE($\{s_1, \ldots, s_n\}$):
    **if** $n \leq 1$ **then**
        **return** $1.0$;
        ;                            `// Only one step ⇒ perfectly uniform`
    Clamp all $s_i \geq 0$;
    $T \leftarrow \sum_i s_i$;
    **if** $T \leq 0$ **then**
        **return** $1.0$;
    $p_i \leftarrow s_i/T$ ;              `// Normalize to probability distribution`
    $H \leftarrow -\sum_i p_i \cdot \log(p_i + \epsilon)$ ;       `// Entropy, ` $\epsilon = 10^{-12}$
    $H_{\max} \leftarrow \log(n)$;
    **if** $H_{\max} = 0$ **then**
        **return** $1.0$;
    **return** $H/H_{\max}$ ;              `// Uniformity score in ` $[0, 1]$
**Function** DETERMINEEVICTIONPERCENTAGE($u, \tau$):
    **if** $u > 0.8$ **then**
        **return** $0.0$ ;         `// High uniformity: keep all steps`
    $e \leftarrow \tau \cdot (1 - u)$ ;       `// Scale eviction by non-uniformity`
    **return** $\min(e, 0.8)$ ;         `// Cap eviction at 80%`
$u \leftarrow$ CALCULATEUNIFORMITYSCORE($\{s_1, \ldots, s_n\}$);
$e \leftarrow$ DETERMINEEVICTIONPERCENTAGE($u, \tau$);

---

## A.9 GRPO DETAILS

For each question $q$, a group of responses $\{y^1, y^2, \ldots, y^N\}$ is sampled from the old policy $\pi_{\text{old}}$, and the policy model $\pi_\theta$ is optimized by maximizing the following GRPO objective.

$$\mathcal{J}_{\text{GRPO}}(\theta) = \frac{1}{N} \sum_{i=1}^{N} \frac{1}{|y^i|} \sum_{t=1}^{|y^i|} \min \left[ \frac{\pi_\theta(y^i(t)|y^i_{<t})}{\pi_{\text{old}}(y^i(t)|y^i_{<t})} \hat{A}_{i,t}, \text{clip} \left( \frac{\pi_\theta(y^i(t)|y^i_{<t})}{\pi_{\text{old}}(y^i(t)|y^i_{<t})}, 1 - \varepsilon, 1 + \varepsilon \right) \hat{A}_{i,t} \right],$$

where $\varepsilon$ is the clipping range hyperparameter, and $\hat{A}_{i,t}$ represents the advantage, computed based on the relative verifiable outcome based rewards of outputs within each group.

## A.10 EXPERIMENTAL DETAILS

We adopt verl (Sheng et al., 2024) as the training framework.

### A.10.1 HYPERPARAMETERS

Table 16: Hyperparameters used for training, evaluation, and difficulty calibration.

| Category | Hyperparameter | Value |
|---|---|---|
| Training | Number of rollouts | 8 |
| | Temperature | 1.0 |
| | top_p | 1.0 |
| | top_k | -1.0 |
| | Max response length | 10k |
| | clip_ratio_low | 0.20 |
| | clip_ratio_high | 0.28 |
| | kl_loss_coef | 0.001 |
| | Learning rate (LR) | 1e-6 |
| Evaluation | Number of rollouts | 8 |
| | Temperature | 1.0 |
| | top_p | 1.0 |
| | top_k | -1.0 |
| | Max response length | 10k |
| | N | 5 |
| Difficulty Calibration | Hard | 0.20 |
| | Medium | 0.40 |
| | Easy | 0.60 |

### A.10.2 TRAINING REWARD

To ensure a high weight on correctness relative to other components, we assign a **correctness reward** of $+4$ if the final answer is correct and 0 otherwise. The **format reward** ranges from 0 to 1: a score of 0.5 is given for the presence of the `<think>` and `</think>` tokens, and an additional 0.5 is awarded if every reasoning trajectory is properly enclosed within these tokens in the correct order. The **length reward** ranges from 0 to 2. The overall reward is computed as the sum of these components:

$$\text{Total Reward} = \text{Correctness Reward} + \text{Format Reward} + \text{Length Reward}.$$

### A.10.3 EVALUATION METRICS

For each of the dataset we compute the accuracy and the average response length. Specifically for OverthinkingBench we also compute the $\text{AUC}_{\text{OAA}}$. This metric is based on Overthinking-Adjusted Accuracy (OAA), which measures model correctness under a limit on reasoning tokens. For a threshold $t$, it is defined as

$$\text{OAA}_t = \frac{1}{n} \sum_{i=1}^{n} \left( \text{Correctness}_i \cdot \mathbb{I}(\text{ThinkTokens}_i < t) \right),$$

where $\text{Correctness}_i \in \{0, 1\}$ indicates whether the $i$-th response is correct, and $\mathbb{I}(\cdot)$ is the indicator function that enforces the thinking length constraint.

$$\text{AUC}_{\text{OAA}} = \int_0^{t_{\max}} \frac{\text{OAA}_t}{t_{\max}} \, dt \approx \frac{1}{t_{\max}} \sum_{t=0}^{t_{\max}} \text{OAA}_t,$$

where $t_{\max}$ is the maximum number of allowed thinking tokens. Furthermore, following the method from (Aggarwal et al., 2025), to compute the OptimalThinkingBench metric: F1 score we combine the $\text{AUC}_{\text{OAA}}$ from OverthinkingBench and Accuracy ($\text{Acc}_{\text{ut}}$) from UnderthinkingBench into a single F1 score:

$$F1 = 2 \cdot \frac{\text{AUC}_{\text{OAA}} \cdot \text{Acc}_{\text{ut}}}{\text{AUC}_{\text{OAA}} + \text{Acc}_{\text{ut}}} \tag{1}$$

### A.10.4 TRAINING PROMPT

For each questions in the training set, instruction was provided: ``Let's think step by step and output the final answer within \\boxed{}''

### A.10.5 BASELINE PROMPTS

Below we define the instruction that was provided to each baseline model:

- Base Model: ``Let's think step by step and output the final answer within \\boxed{}''
- L1-Max: ``Let's think step by step and output the final answer within \\boxed{}. Think for maximum 10000 tokens.''
- LC-R1: `` Please reason step by step, and put your final answer within \\boxed{}''
- AdaptThink: No prompt, just the question
- TokenSkip: "<|im_start|>system You are a helpful assistant.<|im_end|> <|im_start|>user Please reason step by step, and put your final answer within \boxed{}. question<|eot_id|>0.5<|eot_id|><|im_end|> <|im_start|>assistant"

## A.11 COMPUTE USED

All training was done on 4*A100 (80GB).

## A.12 EXAMPLE

**Data Source:** AMC

**Input:**

> Maureen is keeping track of the mean of her quiz scores this semester. If Maureen scores an 11 on the next quiz, her mean will increase by 1. If she scores an 11 on each of the next three quizzes, her mean will increase by 2. What is the mean of her quiz scores currently? Let's think step by step and output the final answer within boxed{}

**Response (TRAAC):**

> \<think\>\\nOkay, let's try to solve this problem. So, Maureen's current mean is what we need to find. The problem gives two scenarios. Let's break it down. First, if she scores an 11 on the next quiz, her mean increases by1. Second, if she gets11 on each of the next three quizzes, the mean increases by2. Hmm. Let's think about how means work. Let's say currently she has a mean of M. Let's say she has n quizzes so far. The total sum of her current scores would be

M multiplied by n. Wait, when she takes another quiz, the number of quizzes becomes n+1. For the first scenario, adding an11would make the new mean be M \+1. So the new total would be (current sum) \+11. And that divided by (n+1) equals the new mean. So (current sum) \+11 \= (M+1) \* (n+1). But the current sum is M \*n. So substituting, M\*n \+11 \= (M+1)(n+1). Let's expand the right side: M\*n \+M \+n \+1. So the equation would be M\*n \+11 \= M\*n \+M \+n \+1. Subtract M\*n from both sides:11 \= M \+n \+1. So M \+n equals10? Because11-1 is10. So M \+n=10? That's equation one.\\n\\nThen the second scenario: if she scores11 on each of thenext three quizzes. So adding three quizzes, each11. The new mean would be current mean plus2. Let's think. The current sum is M\*n. After three more quizzes, the total sum is M\*n \+3\*11. The number of quizzes is n+3. The new mean is M+2. So (M\*n \+33) / (n+3) \= M+2. Let's multiply both sides by(n+3): M\*n \+33 \= (M+2)(n+3). Let's expand the right side: M\*n \+3M \+2n \+6. So the equation becomes M\*n \+33 \= M\*n \+3M \+2n \+6. Subtract M\*n from both sides:33=3M \+2n \+6. Subtract6:27=3M \+2n. So that's the second equation.\\n\\nSo from the first scenario, we had M \+n=10. Let's call that equation1. And the second gives3M \+2n=27? Equation2. So we need to solve these two equations. Let's see. From equation1, n is10-M. Substitute into equation2. So3M \+2\*(10-M) \=27. Let's calculate that:3M \+20-2M \=27. So (3M-2M) is M. So M \+20=27. Therefore M is7? Because27-20is7. So the current mean is7? Let's check. \\n\\nIf M is7, then n is10-M which is3. So currently, she has3 quizzes. Current total is7\*3=21. First scenario: adding11. New total is21+11=32. Number of quizzes is4. Mean would be32/4=8. Which is7+1. Correct. For the second scenario, adding three11s. The total would be21+33=54. Number of quizzes is3+3=6.54/6 is9. The current mean was7, so7+2=9. Yes that's right. So the current mean is7?\\n\</think\>\\n\\nTo determine the current mean of Maureen's quiz scores, we analyze the two given scenarios mathematically.\\n\\n---\\n\\n\#\#\# Step 1: Define Variables\\n\\nLet:\\n- $ M $ \= current mean of quiz scores\\n- $ n $ \= number of quizzes Maureen has taken so far\\n\\nThen the total of her current scores is $ M \\\times n $.\\n\\n---\\n\\n\#\#\# Step 2: First Scenario\\n\\nIf Maureen scores an 11 on the next quiz, her mean increases by 1.\\n\\nThe new total becomes $ M \\\times n \+ 11 $, and the number of quizzes is $ n \+ 1 $. The new mean is $ M \+ 1 $. So:\\n\\n$$\\n\\\\frac{Mn \+ 11}{n \+ 1} \= M \+ 1\\n$$\\n\\nMultiply both sides by $ n \+ 1 $:\\n\\n$$\\nMn \+ 11 \= (M \+ 1)(n \+ 1)\\n$$\\n\\nExpanding the right-hand side:\\n\\n$$\\nMn \+ 11 \= Mn \+ M \+ n \+ 1\\n$$\\n\\nSubtract $ Mn $ from both sides:\\n\\n$$\\n11 \= M \+ n \+ 1 \\\\Rightarrow M \+ n \= 10 \\\\quad \\\\text{(Equation 1)}\\n$$\\n\\n---\\n\\n\#\#\# Step 3: Second Scenario\\n\\nIf she scores 11 on each of the next three quizzes, the mean increases by 2.\\n\\nThe total becomes $ Mn \+ 33 $, and the number of quizzes becomes $ n \+ 3 $. The new mean is $ M \+ 2 $. So:\\n\\n$$\\n\\\\frac{Mn \+ 33}{n \+ 3} \= M \+ 2\\n$$\\n\\nMultiply both sides by $ n \+ 3 $:\\n\\n$$\\nMn \+ 33 \= (M \+ 2)(n \+ 3)\\n$$\\n\\nExpanding the right-hand side:\\n\\n$$\\nMn \+ 33 \= Mn \+ 3M \+ 2n \+ 6\\n$$\\n\\nSubtract $ Mn $ from both sides:\\n\\n$$\\n33 \= 3M \+ 2n \+ 6 \\\\Rightarrow 3M \+ 2n \= 27 \\\\quad \\\\text{(Equation 2)}\\n$$\\n\\n---\\n\\n\#\#\# Step 4: Solve the System of Equations\\n\\nFrom Equation 1:\\n\\n$$\\nM \+ n \= 10 \\\\Rightarrow n \= 10 \- M\\n$$\\n\\nSubstitute into Equation 2:\\n\\n$$\\n3M \+ 2(10 \- M) \= 27 \\\\Rightarrow 3M \+ 20 \- 2M \= 27 \\\\Rightarrow M \+ 20 \= 27 \\\\Rightarrow M \= 7\\n$$\\n\\n---\\n\\n\#\#\# Final Answer\\n\\n$$\\n\\\\boxed{7}\\n$$

**Response (Base Model + CR):**

\<think\>\\nOkay, let me try to figure out this problem. So, Maureen is tracking her mean quiz score. The problem says that if she scores an 11 on the

next quiz, her mean will increase by 1\. Also, if she scores 11 on each of the next three quizzes, her mean will increase by 2\. We need to find her current mean. Hmm, let me break this down step by step.\\n\\nFirst, let me recall what a mean is. The mean is the sum of all the scores divided by the number of scores. So, if I let her current mean be something, maybe I can set up equations based on the information given.\\n\\nLet me denote the current number of quizzes she has taken as $ n $, and her current total score as $ S $. Therefore, her current mean is $ \\\\frac{S}{n} $.\\n\\nNow, the first condition: if she scores an 11 on the next quiz, her mean will increase by 1\. Let's think about that. If she takes one more quiz with a score of 11, then her total score becomes $ S \\+ 11 $, and the number of quizzes becomes $ n \\+ 1 $. The new mean is $ \\\\frac{S \\+ 11}{n \\+ 1} $. According to the problem, this new mean is 1 more than her current mean. So, we can write the equation:\\n\\n$$\\n\\\\frac{S \\+ 11}{n \\+ 1} \\= \\\\frac{S}{n} \\+ 1\\n$$\\n\\nOkay, that seems like a good starting point. Let me write that down.\\n\\nSimilarly, the second condition: if she scores an 11 on each of the next three quizzes, her mean will increase by 2\. So, taking three more quizzes with 11 each, her total score becomes $ S \\+ 11\\*3 \\= S \\+ 33 $, and the number of quizzes becomes $ n \\+ 3 $. The new mean is $ \\\\frac{S \\+ 33}{n \\+ 3} $, which should be 2 more than her current mean. So, the equation would be:\\n\\n$$\\n\\\\frac{S \\+ 33}{n \\+ 3} \\= \\\\frac{S}{n} \\+ 2\\n$$\\n\\nSo, now we have two equations with two variables, $ S $ and $ n $. Let me write both equations again:\\n\\n1. $ \\\\frac{S \\+ 11}{n \\+ 1} \\= \\\\frac{S}{n} \\+ 1 $\\n2. $ \\\\frac{S \\+ 33}{n \\+ 3} \\= \\\\frac{S}{n} \\+ 2 $\\n\\nI need to solve these equations for $ S $ and $ n $, and then find the current mean $ \\\\frac{S}{n} $.\\n\\nLet me start with the first equation. Let's simplify it.\\n\\nMultiply both sides by $ n(n \\+ 1) $ to eliminate denominators:\\n\\n$ (S \\+ 11) \\\\cdot n \\= (S \\+ n(n \\+ 1)) \\\\cdot 1 $\\n\\nWait, let me check that again. Wait, the right-hand side is $ \\\\frac{S}{n} \\+ 1 $, so when I multiply both sides by $ n(n \\+ 1) $, it becomes:\\n\\nLeft side: $ (S \\+ 11) \\\\cdot n $\\n\\nRight side: $ \\[ \\\\frac{S}{n} \\+ 1 \\] \\\\cdot n(n \\+ 1) $\\n\\nWait, let me do that more carefully:\\n\\nOriginal equation:\\n\\n$ \\\\frac{S \\+ 11}{n \\+ 1} \\= \\\\frac{S}{n} \\+ 1 $\\n\\nMultiply both sides by $ n(n \\+ 1) $:\\n\\nLeft side: $ (S \\+ 11) \\\\cdot n $\\n\\nRight side: $ \\[ \\\\frac{S}{n} \\+ 1 \\] \\\\cdot n(n \\+ 1) \\= \\[ S \\+ n \\] \\\\cdot (n \\+ 1) $\\n\\nWait, because $ \\\\frac{S}{n} \\\\times n(n \\+ 1) \\= S(n \\+ 1) $, and $ 1 \\\\times n(n \\+ 1) \\= n(n \\+ 1) $. Therefore:\\n\\nRight side: $ S(n \\+ 1) \\+ n(n \\+ 1) $\\n\\nTherefore, equation becomes:\\n\\n$ n(S \\+ 11) \\= S(n \\+ 1) \\+ n(n \\+ 1) $\\n\\nLet me expand both sides:\\n\\nLeft side: $ nS \\+ 11n $\\n\\nRight side: $ Sn \\+ S \\+ n^2 \\+ n $\\n\\nNow, subtract left side from both sides:\\n\\n0 \\= (Sn \\+ S \\+ n² \\+ n) \\- (nS \\+ 11n)\\n\\nSimplify:\\n\\nSn \\+ S \\+ n² \\+ n \\- nS \\-11n \\= S \\+ n² \\-10n \\= 0\\n\\nTherefore:\\n\\nS \\+ n² \\-10n \\= 0 → S \\= \\-n² \\+10n\\n\\nSo that's equation 1: S \\= \\-n² \\+10n\\n\\nOkay, now let's do the same for the second equation:\\n\\n$ \\\\frac{S \\+ 33}{n \\+ 3} \\= \\\\frac{S}{n} \\+ 2 $\\n\\nMultiply both sides by $ n(n \\+ 3) $:\\n\\nLeft side: (S \\+ 33) \\* n\\n\\nRight side: \\[ S/n \\+ 2 \\] \\* n(n \\+ 3) \\= \\[ S \\+ 2n \\] \\* (n \\+ 3 )\\n\\nTherefore:\\n\\nLeft side: nS \\+ 33n\\n\\nRight side: (S \\+ 2n)(n \\+ 3) \\= S(n \\+ 3) \\+ 2n(n \\+ 3) \\= Sn \\+ 3S \\+ 2n² \\+ 6n\\n\\nTherefore, equation:\\n\\nnS \\+ 33n \\= Sn \\+ 3S \\+ 2n² \\+ 6n\\n\\nSubtract Sn from both sides:\\n\\n33n \\= 3S \\+ 2n² \\+6n\\n\\nBring all terms to left side:\\n\\n33n \\-6n \\-2n² \\-3S \\= 0 → 27n \\-2n² \\-3S \\= 0\\n\\nRearranged:\\n\\n-2n² \\+27n \\-3S \\= 0\\n\\nMultiply both sides by \\-1:\\n\\n2n² \\-27n \\+3S \\=0\\n\\nSo:\\n\\n3S \\= \\-2n² \\+27n → S \\= (-2n² \\+27n)/3\\n\\nSo now, from equation 1, we have S \\= \\-n² \\+10n, and from equation 2, S \\= (-2n² \\+27n)/3.\\n\\nTherefore, we can set them equal:\\n\\n-n² \\+10n \\= (-2n² \\+27n)/3\\n\\nMultiply both sides by 3:\\n\\n-3n² \\+30n \\=

\-2n² \+27n\\n\\nBring all terms to left side:\\n\\n-3n² \+30n \+2n² \-27n \= 0 → (-n²) \+3n \=0 → \-n² \+3n \=0 → n(-n \+3)=0\\n\\nSo, solutions are n=0 or \-n \+3=0 → n=3.\\n\\nBut n is the number of quizzes, so it can't be zero. Therefore, n=3.\\n\\nTherefore, the current number of quizzes is 3.\\n\\nThen, substitute n=3 into equation 1: S \= \-n² \+10n \= \-9 \+30 \= 21\\n\\nTherefore, current total score S=21, and number of quizzes n=3, so current mean is 21/3=7.\\n\\nWait, so the current mean is 7?\\n\\nBut let me check if this works with the original conditions.\\n\\nFirst condition: if she scores an 11 on the next quiz, the mean increases by 1.\\n\\nCurrent mean is 7, with 3 quizzes.\\n\\nAfter taking 11, total is 21 \+11=32, number of quizzes is 4\. New mean is 32/4=8, which is 7 \+1=8. Correct.\\n\\nSecond condition: if she scores 11 on each of the next three quizzes, then total is 21 \+33=54, number of quizzes is 3 \+3=6. New mean is 54/6=9, which is 7 \+2=9. Correct. Perfect.\\n\\nTherefore, current mean is 7\. So the answer is 7, which is \\\\boxed{7}\\n\\n\*\*Final Answer\*\*\\nThe mean of Maureen's quiz scores currently is \\\\boxed{7}.\\n\</think\>\\n\\nTo determine Maureen's current mean quiz score, we start by modeling the situation mathematically.\\n\\n---\\n\\n\#\#\# Step 1: Define Variables\\n\\nLet:\\n- $ n $ be the current number of quizzes Maureen has taken.\\n- $ S $ be her total score so far.\\n- Her current mean is then $ \\\\frac{S}{n} $.\\n\\n---\\n\\n\#\#\# Step 2: Use Given Conditions to Form Equations\\n\\n\#\#\#\# First Condition:\\nIf Maureen scores an \*\*11\*\* on the next quiz, her mean increases by 1.\\n\\nNew total score: $ S \+ 11 $  \\nNew number of quizzes: $ n \+ 1 $  \\nNew mean: $ \\\\frac{S \+ 11}{n \+ 1} $\\n\\nAccording to the problem:\\n$$\\n\\\\frac{S \+ 11}{n \+ 1} \= \\\\frac{S}{n} \+ 1\\n$$\\n\\nMultiply both sides by $ n(n \+ 1) $ to eliminate denominators:\\n$$\\n(S \+ 11)n \= S(n \+ 1) \+ n(n \+ 1)\\n$$\\n\\nSimplify:\\n$$\\nSn \+ 11n \= Sn \+ S \+ n^2 \+ n\\n\\\\Rightarrow 11n \= S \+ n^2 \+ n\\n\\\\Rightarrow S \= \-n^2 \+ 10n\\n$$\\n\\n\#\#\#\# Second Condition:\\nIf Maureen scores an \*\*11\*\* on \*\*three\*\* quizzes, her mean increases by 2.\\n\\nNew total score: $ S \+ 33 $  \\nNew number of quizzes: $ n \+ 3 $  \\nNew mean: $ \\\\frac{S \+ 33}{n \+ 3} $\\n\\nAccording to the problem:\\n$$\\n\\\\frac{S \+ 33}{n \+ 3} \= \\\\frac{S}{n} \+ 2\\n$$\\n\\nMultiply both sides by $ n(n \+ 3) $:\\n$$\\n(S \+ 33)n \= (S \+ 2n)(n \+ 3)\\n$$\\n\\nSimplify:\\n$$\\nSn \+ 33n \= Sn \+ 3S \+ 2n^2 \+ 6n\\n\\\\Rightarrow 33n \= 3S \+ 2n^2 \+ 6n\\n\\\\Rightarrow 27n \= 3S \+ 2n^2\\n\\\\Rightarrow 3S \= \-2n^2 \+ 27n\\n\\\\Rightarrow S \= \\\\frac{-2n^2 \+ 27n}{3}\\n$$\\n\\n---\\n\\n\#\#\# Step 3: Solve for $ n $ and $ S $\\n\\nFrom the two expressions for $ S $:\\n$$\\n-n^2 \+ 10n \= \\\\frac{-2n^2 \+ 27n}{3}\\n$$\\n\\nMultiply both sides by 3:\\n$$\\n-3n^2 \+ 30n \= \-2n^2 \+ 27n\\n\\\\Rightarrow \-n^2 \+ 3n \= 0\\n\\\\Rightarrow n(n \-3) \= 0\\n$$\\n\\nSince $ n \\\\neq 0 $, we get:\\n$$\\nn \= 3\\n$$\\n\\nSubstitute into $ S \= \-n^2 \+ 10n $:\\n$$\\nS \= \-9 \+ 30 \= 21\\n$$\\n\\n---\\n\\n\#\#\# Step 4: Compute the Current Mean\\n\\n$$\\n\\\\text{Current mean} \= \\\\frac{S}{n} \= \\\\frac{21}{3} \= \\\\boxed{7}\\n$$\\n\\n---\\n\\n\#\#\# Final Answer\\n\\n$$\\nboxed{7}$$

**Ground Truth:**