# OpenReview forum: "Think Right: Learning to Mitigate Under-Over Thinking via Adaptive, Attentive Compression"
_ICLR.cc/2026/Conference — Submitted to ICLR 2026_

### Official Review · Reviewer_DSYx · 2025-10-25

**Soundness:** 2
**Presentation:** 2
**Contribution:** 2
**Rating:** 4
**Confidence:** 4

**Summary:**

It proposes a novel online post-training reinforcement learning (RL) method designed to address under-adaptivity in reasoning language models—i.e., their tendency to either underthink (terminate too early on hard problems) or overthink (generate excessive, redundant steps on easy problems).
Despite being trained only on math data (DAPO-Math-17k), TRAAC generalizes to out-of-distribution non-math tasks (e.g., GPQA-D, BBEH, OptimalThinkingBench), showing consistent gains in both accuracy and efficiency.

TRAAC enables language models to "think right": neither too little nor too much—optimizing both reasoning quality and token efficiency through adaptive, attention-guided compression.

**Strengths:**

1. Difficulty-aware compression: Using rollout pass rates during GRPO to estimate problem difficulty and modulate compression intensity。
2. Self-attention as a salience signal: Leveraging the model’s own attention from the </think> token to identify and prune low-importance reasoning steps—without external annotators or auxiliary models.
3. Uniformity-aware pruning: Introducing a mechanism to reduce compression when attention is diffuse (i.e., when no step clearly dominates), preventing harmful over-pruning.
4. Reducing reasoning length by ~37% while boosting accuracy by 8.4% has direct implications for cost, latency, and scalability in real-world deployments.

**Weaknesses:**

1. The results are mainly on small-scale model size such as qwen3-4b/deepseek-r1-7b, it lacks of large-scale model-size exp.
2. Compared to [1], the length-reduction is not as good as [1].
3. The performance of deepseek-r1-7b on aime24 seems lower compared to [2][1] which is 55.4, while the results on aime24 on this work is only 33.71. It is confusing.



[1] DLER: Doing Length pEnalty Right - Incentivizing More Intelligence per Token via Reinforcement Learning
[2] DeepSeek-R1: Incentivizing Reasoning Capability in LLMs via Reinforcement Learning

**Questions:**

1. The performance of deepseek-r1-7b on aime24 seems lower compared to [2][1] which is 55.4, while the results on aime24 on this work is only 33.71. It is confusing.
2. The training reward is the summarization of Correctness Reward,  Format Reward and Length Reward. Did you try to do the ratio search to balance the different aspect and which one is important for length-reduction?






[1] DLER: Doing Length pEnalty Right - Incentivizing More Intelligence per Token via Reinforcement Learning
[2] DeepSeek-R1: Incentivizing Reasoning Capability in LLMs via Reinforcement Learning

---

> ### Author Response · Authors · 2025-11-21
> **Response to Reviewer DSYx**
>
> We thank the reviewers for their careful evaluation and appreciate the recognition of our method’s central contributions, including “using rollout pass rates during GRPO to estimate problem difficulty and modulate compression intensity,” as well as the use of “the model’s own self-attention to identify and prune low-importance reasoning steps without external annotators or auxiliary models.” We are also encouraged by the reviewers’ appreciation of our uniformity-aware pruning mechanism and the practical impact of “reducing reasoning length by ~37% while boosting accuracy by 8.4%,” which directly benefits cost and scalability.
>
> Below, we address each of the raised questions in detail.
>
> ---
> > TRAAC on diverse architecture
>
> To show TRAAC’s robustness across different model architectures, we additionally trained TRAAC on Microsoft/Phi-4-mini-reasoning model, (4B model). The table below presents results comparing the TRAAC-trained Phi-4-mini-reasoning model with the base model. Overall, TRAAC (Phi-4-mini) across AMC and GPQA achieves an performance improvement of 1.71% and delivers a 21% boost in efficiency relative to the base model.
>
> | Method (Phi 4 \- mini) | AMC |  | GPQA |  |
> | :---- | :---- | :---- | :---- | :---- |
> |  | Acc. (%) | Len. (k) | Acc. (%) | Len. (k) |
> | Base model (rerun) | 68.6 | 5.8 | 42.6 | 7.7 |
> | Base Model \+ CR | 66.26 | 5.7 | 38 | 7.6 |
> | Base Model \+ CR \+ LR | 66.26 | 5.5 | 42.13 | 7.2 |
> | TRAAC | 70.12 | 5 | 44.5 | 5.6 |
>
> ---
> > Confusion regarding Deepseek-R1-Qwen-7b performance
>
> The results reported for Deepseek-R1-Qwen-7B in \[2\] are based on the AIME24 dataset, using a generation length of 32k tokens and 64 sampled responses per query to compute pass@1 accuracy. However, our evaluation setup is different, we evaluate models on all questions combined from AIME 22, AIME 23, and AIME 24, and use a maximum generation length of 10k tokens and report the mean accuracy over 5 runs. Therefore there is a difference between the reported numbers. To verify the correctness of our evaluation script, we replicated the exact setup used in \[2\] and obtained an accuracy of 53.8%, which is very close to the reported value of 55.5% in \[2\].
>
> ---
> > Comparison with DLER Baselines
>
> DLER (Doing Length pEnalty Right) \[1\] paper works on a similar problem area and achieves both accuracy and efficiency performance boost. However we identify this as a concurrent work, as the paper was released on 16th Oct 2025, which is after ICLR submission deadline (24th September 2025\) . As a result, it was not possible to include a comparison between TRAAC and DLER in our submission.
>
> ---
> > Analysis of different Reward values
>
> During TRAAC training, we used a correctness reward of \+4/0, format reward of \+1/0 and length reward of \+2/0. The length reward is also kept adaptive rollouts receive a positive length reward only if the final answer is correct; otherwise, all length rewards are set to 0\. The intuition is to always prioritize correctness over efficiency – hence the correctness reward is significantly larger than the others. To study the impact of this reward design, we conducted two additional ablations:
>
> 1. Correctness reward reduced to \+1, with length reward still adaptive to final-answer correctness.
> 2. Correctness reward reduced to \+1, with length reward made independent of final-answer correctness.
>
>
> | Method | AIME |  | AMC |  | GPQA |  |
> | :---- | :---- | :---- | :---- | :---- | :---- | :---- |
> |  | Acc. | Len. | Acc. | Len. | Acc. | Len. |
> | TRAAC\_{reduced correctness} | 29.96% | 6.2k | 71.32% | 3.9k | 47.7k | 3.6k |
> | TRAAC\_{non adaptive} | 5.33% | 0.6k | 34.87% | 0.7k | 29.79k | 0.5k |
> | TRAAC | 45.45% | 6.7k | 79.52% | 4.2k | 47.21% | 4.2k |
>
> The table above compares these variants with the full TRAAC setup. Reducing the correctness reward maintains accuracy on AMC and GPQA, but causes a substantial drop on AIME. Removing the dependency of the length reward on correctness leads the model to exploit the reward by simply shortening its outputs, resulting in severe performance degradation across all datasets.
>
> \[1\] DLER: Doing Length pEnalty Right \- Incentivizing More Intelligence per Token via Reinforcement Learning
> \[2\] DeepSeek-R1: Incentivizing Reasoning Capability in LLMs via Reinforcement Learning
>
> We have added the above results to section 4.2 (Highlighted in violet color).

---

> > ### Comment · Reviewer_DSYx · 2025-11-25
> >
> > Thank you for your reply.
> >
> > I believe it would be better to adhere to the official evaluation setting to ensure a fair comparison across all methods. Using slightly different evaluation protocols can be confusing. That said, I feel the rebuttal has adequately addressed my concerns.
> >
> > Reward shaping plays a crucial role in RL training, and the results under various reward-shaping settings clearly demonstrate the superiority of the proposed method.
> >
> > I will raise my score.

---

> > > ### Author Response · Authors · 2025-11-25
> > > **Response to Reviewer DSYx**
> > >
> > > Thanks for raising your score and for your continued engagement, and for highlighting the crucial role of our reward-shaping -- given additional time, we will add evaluations on the full 32k length in our final/camera-ready version.

---

### Official Review · Reviewer_Ty8v · 2025-10-28

**Soundness:** 3
**Presentation:** 4
**Contribution:** 2
**Rating:** 2
**Confidence:** 4

**Summary:**

The paper presents TRAAC, a reinforcement learning framework designed to mitigate under-adaptivity in the reasoning process of large language models.

TRAAC introduces an adaptive attentive compression module that leverages the attention pattern surrounding the `</think>` token and integrates difficulty-level calibration. The method is further enhanced by a reward system that jointly considers correctness, output format, and reasoning length.

Experimental results demonstrate that TRAAC improves both performance and efficiency while maintaining strong generalization capability across different domains.

**Strengths:**

1. The paper proposes the TRAAC method, offering a novel and well-motivated approach for enhancing the fine-tuning process of large reasoning models.
2. The experimental evaluation includes a comprehensive set of recent and competitive baselines, effectively validating the robustness and effectiveness of the proposed approach.

**Weaknesses:**

1. The approach relies heavily on the model’s inherent reasoning ability, as the training data are generated from the model itself. This dependence makes it difficult to eliminate biases or artifacts inherited from previous training phases, potentially limiting the robustness of the learned behavior.
2. The experiments are conducted only on small models with similar architectures (DeepSeek-Qwen-7B and Qwen3-4B), both of which are Qwen-based and up to 7B scale. This narrow model selection leaves the reliability and scalability of the TRAAC method on larger or architecturally diverse models unverified.

**Questions:**

1. What is the motivation for adopting a reinforcement learning (RL) framework in this context? Using truncated reasoning traces for supervised fine-tuning (SFT) appears to be a more intuitive approach—could the authors clarify the specific advantages of RL in this setting?
2. For models without inherent reasoning capabilities, is it possible to directly train reasoning ability by leveraging reasoning traces generated by other, more capable models?
3. In the attention-based compression stage, the selection is guided solely by the attention pattern around the `</think>` token. Is this approach sufficient to capture all relevant dependencies? Why not consider the full-context attention following the `</think>` token for a potentially more comprehensive compression strategy?

---

> ### Author Response · Authors · 2025-11-21
> **Response to Reviewer Ty8v [Part 1]**
>
> We thank the reviewers for their thoughtful assessment and appreciate the recognition that TRAAC “offers a novel and well-motivated approach for enhancing the fine-tuning process of large reasoning models,” and that our experimental study “includes a comprehensive set of recent and competitive baselines, effectively validating the robustness and effectiveness of the proposed approach.”
>
> Below, we address each of the raised questions in detail.
>
> > Approach relies on the model’s inherent capabilities
>
> Choosing RL rather than SFT to teach adaptive compression is mainly due to two broad reasons:
>
> * Prior work has shown that RL-based methods have shown much better generalization performance as compared to SFT \[1\]. In our current paper SFT based baseline method, TokenSkip, has shown poorer performance as compared to TRAAC, indicating that simply applying SFT to compressed outputs is insufficient.
> * Our method, TRAAC, relies on adaptively compressing reasoning trajectories based on the difficulty of the problem, where a difficulty signal that is itself tightly coupled to the model’s capabilities. Because TRAAC learns this adaptivity during training, it is naturally suited for an online training setup: estimating difficulty becomes easier and automatically adjusts as the model’s reasoning abilities improve over time.
>
>
> **SFT on attention-based compressed trajectories:**
> For a direct comparison, we also train an SFT model using data generated using TRAAC attention-based compressed rollouts. For generating dataset we use a larger model Qwen/Qwen3-32B, for generating the dataset and apply attention-based pruning and use a smaller model (Qwen/Qwen3-4B) on the resulting 1.4k compressed rollouts. The table below summarizes the performance of SFT performance compared to base and TRAAC. The SFT model shows on par performance with the base model and an efficiency boost across the three benchmarks. However TRAAC shows both a boost in performance as well as efficiency when compared to the SFT model.
>
> | Method | AIME |  | AMC |  | GPQA |  |
> | :---- | ----- | ----- | ----- | ----- | ----- | ----- |
> |  | Acc. | Len. | Acc. | Len. | Acc. | Len. |
> | TokenSkip | 5.84 | 9.6 | 27.71 | 8.7 | 32.32 | 7.8 |
> | Base model | 27.64	 | 9.2	 | 68.19	 | 7.0	 | 45.18	 | 7.6 |
> | SFT | 26.06 | 8.8 | 59.51 | 6.6 | 42.0 | 6.9 |
> | TRAAC | 45.45 | 6.7 | 79.52 | 4.2 | 47.21 | 4.2 |
>
> \[1\]: Dubois, Yann, et al. "Length-controlled alpacaeval: A simple way to debias automatic evaluators." arXiv preprint arXiv:2404.04475 (2024).
>
> We have added the above results to Appendix A.4 (Highlighted in violet color).
>
> ---
>
> > TRAAC on diverse architecture
>
> To show TRAAC’s robustness across different model architecture, we additionally trained TRAAC on Microsoft/Phi-4-mini-reasoning model, (4B model). The table below presents results comparing the TRAAC-trained Phi-4-mini-reasoning model with the (i) Base Model; (ii): Base Model \+ CR: The base model trained with GRPO using only the correctness reward (iii) Base model \+ CR \+ LR: The base model trained with GRPO using both correctness and length rewards, but without difficulty-level calibration. Overall, TRAAC (Phi-4-mini-reasoning) across AMC and GPQA achieves an performance improvement of 1.71% and delivers a 21% boost in efficiency relative to the base model.
>
> | Method | AMC |  | GPQA |  |
> | :---- | :---- | :---- | :---- | :---- |
> |  | Acc. (%) | Len. (k) | Acc. (%) | Len. (k) |
> | Base model | 68.6 | 5.8 | 42.6 | 7.7 |
> | Base Model \+ CR | 66.26 | 5.7 | 38 | 7.6 |
> | Base Model \+ CR \+ LR | 66.26 | 5.5 | 42.13 | 7.2 |
> | TRAAC | 70.12 | 5 | 44.5 | 5.6 |

---

> ### Author Response · Authors · 2025-11-21
> **Response to Reviewer Ty8v [Part 2]**
>
> > Attention Score capturing from \</think\> tokens
>
> During generation from a reasoning model, the \</think\> token marks the end of the reasoning process and the answer tokens generated after \</think\> include key summarized conclusions. Prior work \[1\] shows that \</think\> strongly attends to the critical reasoning steps needed to derive the final answer. To verify that attention score from \</think\> is more effective in practice, we ran an additional ablation taking into consideration the full context along with the answer tokens generated after \</think\> token for attention computation. Based on these attention scores, low-scoring steps were removed from the reasoning trajectory. The table below compares the performance of using attention over the full rollout versus TRAAC.
>
> | Method | AIME |  | AMC |  | GPQA |  |
> | :---- | :---- | :---- | :---- | :---- | :---- | :---- |
> |  | Acc. | Len. | Acc. | Len. | Acc. | Len. |
> | TRAAC (full attention rollout) | 2.309% | 1.9k | 18.53% | 2.1k | 25.81% | 4.2k |
> | TRAAC | 45.45  | 6.7k | 79.52% | 4.2k | 47.21% | 4.2k |
>
> Accuracy drops sharply across all three datasets when attention is computed over the full rollout. This demonstrates that without relying on the \</think\> token for attention scoring, the model is unable to reliably identify and prune redundant reasoning steps. Especially on AIME and AMC we observe a substantial drop in efficiency, indicating that when attention is computed over the complete rollout, including both the reasoning and final answer, the model struggles to correctly determine which steps are informative and which are unnecessary.
>
> \[1\]: Choi, Daewon, et al. "Think clearly: Improving reasoning via redundant token pruning." arXiv preprint arXiv:2507.08806 (2025).
>
> We have added the above results to Appendix A.6 (Highlighted in violet color).

---

### Official Review · Reviewer_MoTG · 2025-10-31

**Soundness:** 3
**Presentation:** 2
**Contribution:** 2
**Rating:** 4
**Confidence:** 4

**Summary:**

This paper introduces TRAAC (Think Right with Adaptive, Attentive Compression), an online post-training RL method
that leverages the model’s self-attention over a long reasoning trajectory to identify important steps and prune redundant ones.
Trained with the TRAAC algorithm, reasoning models could learn to think adaptively, thereby enabling more efficient reasoning process.

**Strengths:**

1. This paper is well written and easy to follow.
2. This paper works on an important problem (efficient test time scaling).

**Weaknesses:**

1. **Ignored extra computations** The main contribution of this paper is the proposed TRAAC method, which incoporates an attention-based score to the computation of the reward in online RL. Trained with TRAAC, the reasoning model could think more efficiently on token level. However, the added Attention-Based Compression introduces extra computation complexity. There is no discussion about this problems. Additionally, the impact from different types of attention like Multi-head self attention, Grouped-Query Attention seems to be ignored.

2. **Insufficient evaluations** The experients only includes few reasoning benchmarks (AIME, GPQA). I'm concerned about the generalization of this method to other areas. It's not clear that whether the TRAAC method would do harm to some other abilities like conversations and agentic tasks. Adding these discussion will make this paper more reproducible.

**Questions:**

see Weakness.

---

> ### Author Response · Authors · 2025-11-21
> **Response to reviewer MoTG [Part 1]**
>
> We thank the reviewers for their constructive feedback and for recognizing both the importance of efficient test-time scaling and the strengths of our proposed approach.
>
> Below, we address each of the raised questions in detail.
>
>
> > Computational cost analysis of training TRAAC
>
> To understand the computational cost of training TRAAC vs other RL-based methods (L1-max, AdaptThink), we compare TRAAC with RL baselines on training time and FLOPs.
>
> **Training Time**
> The GRPO algorithm mainly consists of three stages: (i) Rollout: LLM produces multiple responses for a given prompt; (ii) Scoring: A scalar reward is assigned to each response; (iii) Optimising policy: Update the LLM by optimising the total objective. Since we use the math-verify library, a rule-based expression system that does not require additional LLM calls for reward computation, the cost of scoring is negligible. The table below reports the breakdown of training time for TRAAC compared to RL baselines. For each method, we show the time (in seconds) required for the first training step, split into rollout time, policy optimisation time, and total time.
>
> 1) Base Model \+ CR: The base model trained with GRPO using only the correctness reward
> 2) Base model \+ CR \+ LR: The base model trained with GRPO using both correctness and length rewards, but without difficulty-level calibration
>
> Other RL baselines like L1-Max, AdaptThink, are also variants of Base model \+ CR with additional length reward computation.
>
> | Method | Rollout (sec) | Optimise Policy (sec) | Total Time (sec) | Hardware |
> | :---- | :---- | :---- | :---- | :---- |
> | Base Model \+ CR | 250 | 87.5 | 397.5 | H100 |
> | Base model \+ CR \+ LR | 222 | 88 | 375 | H100 |
> | TRAAC | 418 | 88 | 583 | H100 |
>
> ---
> **FLOPs**
> The majority of the difference between TRAAC vs. other RL-based methods lies in the rollout strategy used. TRAAC rollout consists of (i) Generation: Generation of initial reasoning steps; (ii) Attention-based compression: The reasoning trajectory is compressed by calculating the attention score of each reasoning step; (iii) Answer generation: The answer is generated based on the compressed reasoning chain.
>
> The table below compares the FLOPs utilized for generating 20 training examples. Where Base Model + CR: base model trained with GRPO using only the correctness reward.
>
> | Method          | FLOPs Used                |
> |-----------------|---------------------------|
> | Base Model + CR | $1.65 \times 10^{15}$ FLOPs |
> | TRAAC           | $3.84 \times 10^{15}$ FLOPs |
>
>
> Most RL baselines only perform the initial generation step. In contrast, TRAAC adds an additional attention-computation stage, yet keeps the overall FLOPs in the same order of magnitude – while producing higher-quality reasoning trajectories.
>
> Even though TRAAC incurs an increase in training time and FLOPs for the initial batches, mainly due to its multi-stage generation, the overhead is not substantial, and moreover the during inference we make the model more efficiency, effectively reducing the cost during test time.
>
> We have added the above results to Appendix A.2 (Highlighted in violet color).

---

> ### Author Response · Authors · 2025-11-21
> **Response to reviewer MoTG [Part 2]**
>
> > Comprehensive Evaluation of TRAAC
>
> In addition to evaluating TRAAC on AIME, AMC, and GPQA benchmarks, we also evaluated TRAAC on other datasets like BBEH and optimalThinkingBench. Although TRAAC was trained exclusively on a math-focused dataset (Dapo-17k), we extended the evaluation to several non-math benchmarks:
>
> 1. GPQA: A MCQ dataset covering physics, biology, and chemistry
> 2. BBEH: A benchmark consisting of novel tasks like BoardgameQA, Causal Understanding, Spatial Reasoning, etc
> 3. OptimalThinkingBench: Include two sub benchmarks: OverthinkingBench, featuring simple queries in 72 domains from OvT-Math, focusing on simple math problems and OvT-General, consisting of general queries across diverse domains, and UnderthinkingBench, containing 11 challenging reasoning tasks from the reasoning gym.
>
> Including diverse test benchmarks allows us to robustly assess the out-of-distribution generalization capability of TRAAC. However, we conduct an additional evaluation on agentic \+ multi-turn benchmark: MINT \[1\], which measures an LLM’s ability to solve complex tasks through multi-turn interactions and tool use.
>
> In MINT we tasks LLMs to solve different tasks with different interaction limits k ∈ {1, 2, 3, 4, 5} without natural language feedback, and quantify LLMs’ tool-augmented task-solving capability through (1) absolute performance SR (success rate), (2) Average response length
>
> The table below compares the base model (Qwen3-4B) and TRAAC across these metrics for three task categories – code generation, decision making, and reasoning.
>
> | Task | Method | k |  |  |  |  |  |  |  |  |  |
> | :---- | :---- | ----- | ----- | ----- | ----- | ----- | ----- | ----- | ----- | ----- | ----- |
> |  |  | 1 |  | 2 |  | 3 |  | 4 |  | 5 |  |
> |  |  | SR (%) | Len (k) | SR (%) | Len (k) | SR (%) | Len (k) | SR (%) | Len (k) | SR (%) | Len (k) |
> | code\_generation | Base Model | 0.74 | 0.5 | 58.09 | 3.7 | 58.09 | 4.8 | 59.56 | 5.7 | 59.56 | 7.1 |
> |  | TRAAC | 49.26 | 1.7 | 58.82 | 2.6 | 56.62 | 3.3 | 59.56 | 3.5 | 58.82 | 3.8 |
> | decision\_making | Base Model | 0.00 | 0.5 | 11.19 | 2.4 | 17.16 | 2.5 | 30.6 | 2.5 | 33.58 | 3.0 |
> |  | TRAAC | 0.00 | 0.5 | 8.21 | 1.0 | 21.64 | 1.2 | 35.07 | 1.3 | 40.3 | 1.6 |
> | reasoning | Base Model | 19.94 | 0.5 | 76.58 | 1.9 | 80.38 | 2.2 | 79.75 | 2.5 | 79.75 | 2.4 |
> |  | TRAAC | 66.46 | 0.8 | 76.9 | 1.1 | 81.65 | 1.2 | 79.11 | 1.3 | 81.96 | 1.2 |
> | avg\_micro | Base Model | 10.92 | 0.5 | 57.34 | 2.4 | 60.75 | 2.9 | 63.82 | 3.2 | 64.51 | 3.6 |
> |  | TRAAC | 47.27 | 1.0 | 57 | 1.4 | 62.12 | 1.7 | 64.51 | 1.8 | 67.06 | 1.9 |
>
> When looking at the avg\_micro across the interaction limit k ∈ {1, 2, 3, 4, 5}, TRAAC performs on matches or exceeds the base model, with an average success rate increase of 8.12% and also a reduction in response length by 38.3%. This demonstrates that TRAAC not only improves performance on agentic, multi-turn tasks but also makes the model more efficient in its interactions.
>
> \[1\]: Wang, Xingyao, et al. "Mint: Evaluating llms in multi-turn interaction with tools and language feedback." arXiv preprint arXiv:2309.10691 (2023).
>
> We have added the above results to Appendix A.3 (Highlighted in violet color).

---

> ### Comment · Reviewer_MoTG · 2025-11-26
> **Response to the Author**
>
> Dear Authors,
>
> I have carefully read your rebuttal. My concern about the generalization of the method has been well addressed. But the overhead (FLOPs) of TRAAC on train time is 2x than Base Model + CR. It basically uses extra train time to achieve lower test time consumption. So I decide to maintain my score.
>
> Best,
>
> Reviewer MoTG

---

> > ### Author Response · Authors · 2025-11-29
> > **Response to reviewer MoTG**
> >
> > We thank the reviewer for participating in the discussion. Below, we address the additional questions raised in detail.
> >
> > > Computational cost analysis (Training vs Inference)
> >
> > We agree that TRAAC utilises additional FLOP and slightly higher step time during its initial training steps; however, this overhead quickly amortises as the training progresses. As the model learns to shorten its generated reasoning traces, its computational cost – including both FLOPs and time per step steadily decreases. To show this empirically, we have added a new figure in the revised Appendix (A.2.3) showing the time-per-step curve for training DeepSeek-Qwen-7B. As illustrated in the figure, TRAAC begins with a higher step time compared to the Base Model \+ CR baseline, but the gap closes rapidly. Around mid-training, the two curves match closely, and in later steps, TRAAC consistently becomes more efficient – ultimately achieving a lower step time than the baseline.
> >
> > This confirms that while TRAAC introduces an initial overhead, its adaptive reduction of reasoning length leads to substantial efficiency gains, resulting in lower computation over the course of training.
> >
> > To highlight the advantage gained from training TRAAC in terms of FLOPs we calculated the average FLOPs required during inference. We calculated the total amount of FLOPs for generating 80 examples from AMC dataset. The below table show the total amount FLOPs required for both TRAAC and Base Model \+ CR.
> >
> > | Method | Total FLOPs (80 questions) | Average FLOPs per question |
> > | :---- | :---- | :---- |
> > | Base Model \+ CR (Qwen3-4B) | 3.7×1015 FLOPs | 4.6×1013 FLOPs  |
> > | TRAAC (Qwen3-4B) | 2.7×1015 FLOPs | 3.3×1013 FLOPs  |
> >
> > TRAAC yields a substantial reduction in inference compute. As shown above, TRAAC requires 2.7 × 1015 FLOPs to answer 80 AMC questions, compared to 3.7 × 1015 FLOPs for the Base Model \+ CR. This corresponds to a 27.3% reduction in inference FLOPs, or a savings of 1.2664 × 1013 FLOPs per question. Although TRAAC incurs additional compute during the beginning of the training, and quickly amortises as the training progresses. This cost is balanced quickly at inference time.
> >
> > Therefore, with the above two experiments, we show that TAAC not only amortises quickly during inference but also that it, in fact, makes training more efficient within 100 steps.
> >
> > Moreover, several prior works have been done where additional computation is introduced during training, in order to reduce inference-time cost or improve accuracy:
> >
> > MAGDI \[1\] (ICML 2024\) employs structured distillation through multiple rounds of interactions with a large teacher model. Although this substantially increases training-time computation, it enables smaller models to acquire stronger reasoning skills.
> > C3oT \[2\] (AAAI 2025\) leverages an external CoT–compression model (GPT-4) to rewrite long reasoning chains into shorter but semantically equivalent ones. This requires costly teacher queries during training, but results in significantly shorter and more efficient reasoning at inference.
> >  LC-R1 \[3\] introduces a post-training GRPO-based method that uses a separate finetuned model to identify and extract the minimal necessary prefix of a reasoning trajectory, again incurring additional computation during training to reduce inference-time reasoning length.
> >
> > These works collectively demonstrate a well-established pattern in the literature: extra computation during training is a reasonable and widely adopted strategy when it leads to meaningful gains in inference-time efficiency or accuracy. TRAAC is aligned with this paradigm. While it uses additional compute in the early training stages, it directly results in both improved task performance and reduced inference-time cost through adaptive reasoning-length reduction.
> >
> > We hope this addresses your concerns, and it would be great if you could revisit the scores. We are happy to answer any follow-up questions you may have.
> >
> > Thanks
> >
> > [1]: Chen, Justin Chih-Yao, et al. "Magdi: Structured distillation of multi-agent interaction graphs improves reasoning in smaller language models." ICML 2024\.
> > [2]: Kang, Yu, et al. "C3ot: Generating shorter chain-of-thought without compromising effectiveness." Proceedings of the AAAI 2025.
> > [3]:  Cheng, Zhengxiang, et al. "Optimizing Length Compression in Large Reasoning Models." arXiv preprint 2025\.

---

### Official Review · Reviewer_9cDt · 2025-11-01

**Soundness:** 2
**Presentation:** 2
**Contribution:** 2
**Rating:** 4
**Confidence:** 4

**Summary:**

This paper introduces TRAAC, a post-training reinforcement learning method designed to address the problem of under-adaptivity in large language models (LLMs)—the failure to dynamically allocate reasoning effort (thinking length) based on problem difficulty. Under-adaptivity leads to overthinking on simple problems (wasting compute) and underthinking on hard problems (sacrificing accuracy).

**Strengths:**

Explicitly incorporating estimated task difficulty to modulate the compression rate is a clear and direct way to tackle under-adaptivity, which is often only implicitly addressed in prior work.
Results show that TRAAC simultaneously improves accuracy and reduces reasoning length compared to strong baselines, demonstrating effective adaptive reasoning that generalizes to out-of-distribution tasks.

**Weaknesses:**

If the responses are trimmed, they are usually not coherent and readable, which significantly impacts the model's usability and user-friendliness.
The paper relies on the assumption that low attention from </think> implies a reasoning step is redundant. While the results are promising, there is no analysis to validate that the pruned steps are indeed unhelpful or that the attention mechanism is a faithful indicator of reasoning importance.
While TRAAC improves test-time efficiency, the computational cost of the online GRPO training—which involves generating multiple rollouts, computing attention scores for the entire trajectory, and performing compression for each training step—is likely substantial. The paper does not discuss the training efficiency or compare it to the baselines. A discussion of this trade-off (offline training cost vs. online inference savings) is important for a complete picture.

**Questions:**

Could you provide evidence that the attention-based compression is faithful? For a sample of correctly answered problems, can you show that the compressed reasoning trajectory remains logically sound and complete? Conversely, for some failures, could over-compression have removed a critical step? For instance, on a subset of problems, you could use a verifier model or human annotators to score the logical coherence of the full vs. compressed reasoning chains.
What is the comparative computational cost of training TRAAC versus the other RL baselines (L1-Max, AdaptThink)? An estimate of the additional overhead introduced by the attention computation and compression module would help users weigh the benefits against the training costs.

---

> ### Author Response · Authors · 2025-11-21
> **Response to reviewer 9cDt [Part 1]**
>
> We thank the reviewers for providing us with feedback, and acknowledging that “explicitly incorporating estimated task difficulty to modulate the compression rate is a clear and direct way to tackle under-adaptively” and “TRAAC simultaneously improves accuracy and reduces reasoning length compared to strong baselines, demonstrating effective adaptive reasoning that generalizes to out-of-distribution tasks”.
>
> Below, we address each of the raised questions in detail.
>
> > Analysis of pruned reasoning steps based on attention score
>
> **Analysis 1: Use LLM as a judge to calculate coherence between original and pruned reasoning trajectory pairs.**
> To evaluate whether the pruned reasoning trajectories maintain its logical coherence when compared to the original reasoning trajectory, we use LLM as a judge to evaluate the logical coherence and flow between the original and pruned reasoning trajectories. We use meta-llama/Llama-3.3-70B-Instruct as a judge to evaluate logical coherence between two different pairs of trajectories:
> (1): TRAAC’s attention-based trajectory with original reasoning trajectory
> (2): Randomly pruned trajectories with the original reasoning trajectory, where in a randomly pruned trajectory, randomly selected reasoning steps are removed.
>
> Evaluation was done on 200 pairs of trajectories and was assigned a score between 1 to 5\. The table below shows the two results:
>
> | Method | Score |
> | :---- | :---- |
> | Random Pruning | 3.35 |
> | Attention-Based Pruning | 3.71 |
>
> Results show an increase of 0.36 when comparing TRAAC’s attention-based pruning with random pruning, which highlights that attention is able to correctly identify which reasoning steps are redundant and effectively prune them.
>
> ---
> **Analysis 2: Replacing attention-based compression with confidence-based**
> To help understand the efficiency of the adaptive, attentive compression module, we replace the attention-based compression with random step compression or confidence-based compression. At each training step, instead of using attention as a metric, reasoning steps are pruned either randomly or steps with the least confidence. The table below compares TRAAC (Qwen3-4B) with random steps and least confidence. Relative to TRAAC, random step pruning shows an average of 11% accuracy drop, and similarly, pruning
> the least confidence steps leads to a 7.25% accuracy drop. This highlights the efficacy of using attention-based compression in TRAAC.
>
> | Method | AIME |  | AMC |  | GPQA |  |
> | :---- | :---- | :---- | :---- | :---- | :---- | :---- |
> |  | Acc. | Len. | Acc. | Len. | Acc. | Len. |
> | Random Steps | 29.54 | 6.5 | 66.74 | 4.1 | 42.94 | 3.2 |
> | Least Confidence | 32.35 | 5.8 | 71.08 | 3.4  | 47 | 3.0 |
> | TRAAC | 45.45 | 6.7 | 79.52 | 4.2 | 47.2 | 4.2 |
>
> The above analysis has been added in section 4.2 (highlighted in violet color)

---

> ### Author Response · Authors · 2025-11-21
> **Response to reviewer 9cDt [Part 2]**
>
> > Computational cost analysis of training TRAAC
>
> To understand the computational cost of training TRAAC vs other RL-based methods (L1-max, AdaptThink), we compare TRAAC with RL baselines on training time and FLOPs.
>
> **Training Time**
> The GRPO algorithm mainly consists of three stages: (i) Rollout: LLM produces multiple responses for a given prompt; (ii) Scoring: A scalar reward is assigned to each response; (iii) Optimising policy: Update the LLM by optimising the total objective. Since we use the math-verify library, a rule-based expression system that does not require additional LLM calls for reward computation, the cost of scoring is negligible. The table below reports the breakdown of training time for TRAAC compared to RL baselines. For each method, we show the time (in seconds) required for the first training step, split into rollout time, policy optimisation time, and total time.
>
> 1) Base Model \+ CR: The base model trained with GRPO using only the correctness reward
> 2) Base model \+ CR \+ LR: The base model trained with GRPO using both correctness and length rewards, but without difficulty-level calibration
>
> Other RL baselines like L1-Max, AdaptThink, are also variants of Base model \+ CR with additional length reward computation.
>
> | Method | Rollout (sec) | Optimise Policy (sec) | Total Time (sec) | Hardware |
> | :---- | :---- | :---- | :---- | :---- |
> | Base Model \+ CR | 250 | 87.5 | 397.5 | H100 |
> | Base model \+ CR \+ LR | 222 | 88 | 375 | H100 |
> | TRAAC | 418 | 88 | 583 | H100 |
>
> **FLOPs**
> The majority of the difference between TRAAC vs. other RL-based methods lies in the rollout strategy used. TRAAC rollout consists of (i) Generation: Generation of initial reasoning steps; (ii) Attention-based compression: The reasoning trajectory is compressed by calculating the attention score of each reasoning step; (iii) Answer generation: The answer is generated based on the compressed reasoning chain.
>
> The table below compares the FLOPs utilized for generating 20 training examples. Where Base Model + CR: base model trained with GRPO using only the correctness reward.
>
> | Method          | FLOPs Used                |
> |-----------------|---------------------------|
> | Base Model + CR | $1.65 \times 10^{15}$ FLOPs |
> | TRAAC           | $3.84 \times 10^{15}$ FLOPs |
>
> Most RL baselines only perform the initial generation step. In contrast, TRAAC adds an additional attention-computation stage, yet keeps the overall FLOPs in the same order of magnitude, while producing higher-quality reasoning trajectories.
>
> Even though TRAAC incurs an increase in training time and FLOPs for the initial batches, mainly due to its multi-stage generation, the overhead is not substantial, and moreover the during inference we make the model more efficiency, effectively reducing the cost during test time.
>
> We have added the above results to Appendix A.2 (Highlighted in violet color).

---

> > ### Comment · Reviewer_9cDt · 2025-11-26
> >
> > Thank you for your response. I still have concerns about the usability and user-friendliness of a model trained in this manner, so I will maintain my score. In addition, I believe the authors should compare against well-formed reasoning trajectories rather than Random Pruning.

---

> > > ### Author Response · Authors · 2025-11-28
> > > **Response to Reviewer 9cDt**
> > >
> > > We thank the reviewer for participating in the discussion. Below, we address the additional questions raised in detail.
> > >
> > > > Usability and user-friendliness of a model trained in this manner
> > >
> > > TRAAC is an online post-training method that applies adaptive attention-based pruning only during training to reduce redundant reasoning steps and improve trajectory quality. Crucially, the pruning logic is not used at inference, and a TRAAC-trained model operates identically to any standard reasoning mode, i.e during inference, the user only needs to input the question/prompt; there is no additional computation, overhead, interface change, or system-level requirement introduced by our method. As a result, TRAAC’s usability and user-friendliness remain unchanged. We have also added examples of output CoT in the updated manuscript (Appendix A.12), to show that TRAAC reasoning steps follow the same structure and maintain the same quality, and are shorter than the base model.
> > >
> > > > Compare against well-formed reasoning trajectories rather than Random Pruning
> > >
> > > We would like to clarify that our evaluation already includes a comparison between TRAAC and high-quality reasoning trajectories. Specifically, we assessed the logical coherence between original reasoning trajectories with pruned trajectories using Llama-3.3-70B-Instruct as a judge.
> > > In the experiment below, we use the well-formed trajectories and compare them with their corresponding pruned version to calculate logical coherence using LLM as a judge. The table below reports the logical coherence scores, where each row corresponds to the coherence between a pruned trajectory and its original well-formed (unpruned) version. For pruned trajectories, we use TRAAC’s attention-based trajectories and randomly pruned trajectories. Random pruning provides a controlled baseline that evaluates TRAAC’s attention-based pruning to isolate redundant reasoning steps.
> > >
> > >
> > > | Method | Score |
> > > | :---- | :---- |
> > > | Randomly pruned trajectories vs. Original Trajectories | 3.35 |
> > > | TRAAC’s attention-based trajectories vs. Original Trajectories | 3.71 |
> > >
> > > We also compared TRAAC against training models with alternative pruning strategies that use (a) random step removal and (b) confidence-based removal. The baseline in section 4.2 of the paper, Base model \+ CR \+ LR, where the base model is trained with GRPO using only the correctness reward, is a direct comparison with TRAAC, where well-formed reasoning trajectories are used rather than pruned reasoning trajectories.
> > >
> > > The full comparison is shown below. TRAAC consistently outperforms:
> > > (i) training on well-formed (unpruned) trajectories, and
> > > (ii) training on pruned trajectories generated by random or confidence-based removal.
> > >
> > > | Method | AIME |  | AMC |  | GPQA |  |
> > > | :---- | :---- | :---- | :---- | :---- | :---- | :---- |
> > > |  | Acc. | Len. | Acc. | Len. | Acc. | Len. |
> > > | Base Model \+ CR (well-formed reasoning trajectories) | 44.36  | 7.9  | 77.35 | 5.5 | 46.29 | 5.7 |
> > > | Random Steps | 29.54 | 6.5 | 66.74 | 4.1 | 42.94 | 3.2 |
> > > | Least Confidence | 32.35 | 5.8 | 71.08 | 3.4  | 47 | 3.0 |
> > > | TRAAC | 45.45 | 6.7 | 79.52 | 4.2 | 47.2 | 4.2 |
> > >
> > > We hope this addresses your concerns, and it would be great if you could revisit the scores. We are happy to answer any follow-up questions you may have.
> > >
> > > Thanks

---

### Author Response · Authors · 2025-12-02
**Summary of Discussion Period**

We sincerely thank the Area Chair and all reviewers for their time, constructive feedback, and thoughtful engagement throughout the discussion period. We would like to summarise our rebuttal here.

### During the rebuttal, we made targeted revisions addressing all major comments:

* **Experiment on diverse architecture (Raised by: DSYx, Ty8v):** To show TRAAC’s robustness across different model architectures, we additionally trained TRAAC on Microsoft/Phi-4-mini-reasoning model (4B model). TRAAC (Phi-4-mini-reasoning) across AMC and GPQA achieves an performance improvement of 1.71% and delivers a 21% boost in efficiency relative to the base model.
* **Benchmarking on other domains like conversation (Raised by: MoTG):** we conducted an additional evaluation on the agentic \+ multi-turn benchmark: MINT \[1\], which measures an LLM’s ability to solve complex tasks through multi-turn interactions and tool use
* **Why attention-based pruning is effective (Raised by: 9cDt):** We added two analyses (i) Coherence Judging: Using Llama-3.3-70B to score original vs. pruned trajectories, TRAAC’s pruning increased coherence by \+0.36 over random pruning (ii) Alternative Compression: Replacing attention with random or low-confidence pruning caused 11% and 7.25% accuracy drops, respectively – confirming attention is an effective signal.
* **Why attention is computed from \</think\> (Raised by: Ty8v):** Ran an additional ablation taking into consideration the full context along with the answer tokens generated after \</think\> token for attention computation. Moreover, prior work shows that \</think\> strongly attends to the critical reasoning steps needed to derive the final answer.
* **Training-time overhead vs. RL baselines (Raised by: 9cDt, MoTG):** We analysed by calculating the training step and FLOPs of TRAAC vs  RL baselines. TRAAC starts with a slightly higher step time, but as the model learns shorter traces, the cost rapidly decreases. Mid-training onward, TRAAC’s step time matches or is lower than RL baselines. Similarly, for FLOPs, TRAAC’s multi-stage generation causes modest overhead early, but provides substantial inference-time FLOP savings.
* **Why RL and not SFT? (Raised by: Ty8v):** SFT alone (similar to TokenSkip) fails to learn adaptive compression. Still, we trained an SFT baseline using 1.4k TRAAC-pruned rollouts (generated from Qwen-32B). The SFT model shows on par performance with the base model and an efficiency boost across the three benchmarks. However TRAAC shows both a boost in performance as well as efficiency when compared to the SFT model.
* **Reward design questions (Raised by: DSYx):** We ran two ablations showing that reducing the correctness reward hurts performance, and making the length reward independent of correctness causes severe performance collapse, as the model exploits the reward by over-shortening.

### Outcome of the rebuttal:

We would like to highlight that three out of four reviewers participated in the discussion:

* **Reviewer DSYx increased their score from 4 to 6 (25th Nov 2025\)** saying "I feel the rebuttal has adequately addressed my concerns".
* **Reviewer 9cDt** still had a minor comment regarding the usability of the trained model; in response, we clarified that a TRAAC-trained model behaves exactly like any other reasoning model at inference time, with no additional setup or specialised infrastructure required.
* We also successfully addressed **Reviewer MoTG** comments regarding generalisation. In addition, they raised questions about the computational overhead of attention-based compression; to address this, we conducted additional FLOP-usage experiments showing that the extra training cost amortises rapidly as training progresses.
* **Reviewer Ty8v**, who shared comments similar to those of other reviewers (i.e., additional computation analysis and broader benchmark coverage), did not participate in the discussion phase.

### Key strengths identified by the reviewers:

Reviewer 9cDt highlighted that “explicitly incorporating estimated task difficulty to modulate the compression rate is a clear and direct way to tackle under-adaptivity,” improving accuracy and reasoning length. Reviewer MoTG called the paper “well written and easy to follow” and noted it “works on an important problem (efficient test time scaling).” Reviewer Ty8v described TRAAC as “a novel and well-motivated approach for enhancing the fine-tuning process of large reasoning models,” supported by “comprehensive…competitive baselines.” Reviewer DSYx emphasized the technical contributions -- “difficulty aware compression,” “self-attention as a salience signal,” and “uniformity-aware pruning” -- and its impact: “reducing reasoning length by ~37% while boosting accuracy by 8.4%.”

We hope the above summary will help the AC evaluate the paper based on its current improved state and take into consideration the score improvements and discussion we had with the reviewers.

---

### Meta-Review · Area_Chair_1EHe · 2026-01-08

**Summary:**

This paper proposes TRAAC (Think Right with Adaptive, Attentive Compression), an online post-training reinforcement learning method aimed at mitigating under-adaptivity in large language model reasoning, i.e., the tendency to underthink on hard problems and overthink on easy ones. TRAAC leverages the model’s self-attention over long reasoning trajectories to identify salient steps and prune redundant ones, while incorporating difficulty estimation into the reward to modulate compression strength. The method is trained with GRPO and evaluated primarily on math and science reasoning benchmarks (AIME, AMC, GPQA-D, BBEH), reporting simultaneous gains in accuracy and reductions in reasoning length, along with some out-of-distribution generalization.

**Reviewer Concerns:**

A central premise of TRAAC is that self-attention scores provide a faithful signal of reasoning-step importance. While the rebuttal adds auxiliary analyses suggesting attention-based pruning outperforms random alternatives, the evidence remains indirect and model-internal. There is no human- or verifier-grounded validation demonstrating that pruned reasoning chains preserve logical soundness in a principled way. As a result, the method risks optimizing for internal heuristics rather than genuine reasoning quality.

Although TRAAC reduces test-time reasoning length, reviewers consistently raised concerns about the computational overhead introduced by online RL training, including multiple rollouts, full-trajectory attention computation, and compression steps. The rebuttal provides amortized FLOP analyses, but the trade-off remains insufficiently convincing.

Most experiments are conducted on small, closely related model families (Qwen-based and similar architectures, up to 7B). The robustness of TRAAC on larger models, different attention mechanisms, and broader capability suites (e.g., dialogue, tool use, safety-critical reasoning) remains unestablished.

Reviewers viewed the contribution as incremental relative to prior work on reasoning length control, length penalties, and adaptive compute allocation. TRAAC combines attention-based pruning and difficulty-aware rewards in a reasonable but largely heuristic manner, without offering a deeper theoretical justification, general framework, or insight into why this approach should be preferred over simpler alternatives (e.g., length-regularized RL or supervised pruning).

**Reviewer Scores:**

Although one reviewer raised the rating to 6 during the rebuttal period, the majority of reviewers remain negative.

---

### Decision · Program_Chairs · 2026-01-26

Reject